# A blueprint of the topology and mechanics of the human ovary for next-generation bioengineering and diagnosis

Emna Ouni [1], Alexis Peaucelle[2,5], Kalina T. Haas[2,5], Olivier Van Kerk[1], Marie-Madeleine Dolmans[1,3], Timo Tuuri [4], Marjut Otala [4] & Christiani A. Amorim [1✉]

Although the first dissection of the human ovary dates back to the 17[th] century, the biophysical characteristics of the ovarian cell microenvironment are still poorly understood. However, this information is vital to deciphering cellular processes such as proliferation, morphology and differentiation, as well as pathologies like tumor progression, as demonstrated in other biological tissues. Here, we provide the first readout of human ovarian fiber morphology, interstitial and perifollicular fiber orientation, pore geometry, topography and surface roughness, and elastic and viscoelastic properties. By determining differences between healthy prepubertal, reproductive-age, and menopausal ovarian tissue, we unravel and elucidate a unique biophysical phenotype of reproductive-age tissue, bridging biophysics and female fertility. While these data enable to design of more biomimetic scaffolds for the tissue-engineered ovary, our analysis pipeline is applicable for the characterization of other organs in physiological or pathological states to reveal their biophysical markers or design their bioinspired analogs.

[1] Pôle de Recherche en Gynécologie, Institut de Recherche Expérimentale et Clinique, Université Catholique de Louvain, Brussels, Belgium. [2] Institut Jean-Pierre Bourgin, INRAE, AgroParisTech, Université Paris-Saclay, Versailles, France. [3] Gynecology and Andrology Department, Cliniques Universitaires Saint-Luc, Brussels, Belgium. [4] Department of Obstetrics and Gynecology, Helsinki University Hospital, University of Helsinki, Helsinki, Finland. [5] These authors contributed equally: Alexis Peaucelle, Kalina T. Haas. ✉email: christiani.amorim@uclouvain.be

Since 1976, >7 million babies have been born around the world using assisted reproductive technology[1], making medical intervention to aid men and women with infertility routine practice and an increasingly requested technology. Novel fertility restoration strategies, such as ovarian tissue engineering, have emerged in the past decade to respond to patient needs, manage their clinical condition, and offer reliable models for reproductive biology research. Ovarian engineering has relied on synthetic[2] and natural[3–5] hydrogels, decellularized scaffolds[6–8], and microfluidics[9]. However, these techniques were designed based on knowledge acquired from two-dimensional cell culture and animal models. Finding a relevant model for the human ovary is almost impossible as only a few mammals, including several whale species and elephants, have long pre- and post-reproductive stages.

Indeed, our lack of information on the human ovary hampers our ability to mimic the main features of this organ, either in an in vitro model or functional transplantable engineered ovary. The complex composition and hierarchical structure of the extracellular matrix (ECM) complicates the design of truly biomimetic constructs, notably multi-scale ECM organization and architecture, as well as its biomechanical features controlling follicle activation and mechanotransduction[10–12]. Moreover, the ovarian ECM is very dynamic, owing to ovarian tissue changes from prepubertal through reproductive to menopausal age.

In the past, da Vinci drew human corpses in the famed Codex Leicester[13] and depicted the analogy between Earth and the human body in an attempt to understand the human organism. Centuries after his death, modern anatomists were still learning through Leonardo's sketches. This historical fact further highlights the importance of having a reference base upon which we can advance future therapies and technologies.

Today, manufacture of hydrogels with tailored biochemical, architectural, and mechanical features is feasible and constantly developing. Nevertheless, the absence of reliable characterization of the human ovarian ECM is still lacking. Even da Vinci was not able to reproduce the female reproductive tract realistically in his drawings[14].

Here we directly investigate the changes to human ovarian tissue during its lifespan, thus identifying the architectural elements involved in ovarian function. This work reveals unknown facets of the ovarian ECM on multiple scales and their evolution from prepuberty to menopause, aspects that have so far been missing from the engineered ovary. More specifically, all analyses were performed using samples from the ovarian cortical layer, where the ovarian follicle reserve resides. ECM fiber thickness, assembly, orientation, and directionality around follicle borders, pore number and size, topography, and elastic and viscoelastic properties of the human ovarian cortex are all depicted. Our study highlights the impact of local and global fibril orientation and nanoscale and mesoscale ECM porosity on the mechanical properties of ovarian tissue. By affecting chemical and mechanical signaling, these features are critical determinants of ovarian function and should become a blueprint for designing functional artificial ovaries in the near future.

## Results

**Multi-scale fiber thickness and pore properties demonstrate microstructural remodeling with age.** In this study, we define fibrils as individual entities with discrete diameters in the order of several nanometers, whereas fibers are defined as a pack of multiple fibrils that can be assembled into bundles (Fig. 1A). While it is rare to observe isolated fibrils in ovarian cortex (Fig. 1B), we were able to capture the main features of its fibers at ×20,000 magnification and their arrangement into bundles at ×5000 magnification (Fig. 1B, C). After identifying fiber bundles at ×5000, we zoomed in to pinpoint the features of their elementary fibers at ×20,000 (Fig. 2).

The results revealed specific organization of the ovarian ECM at prepuberty, characterized by thin fibers (vs reproductive age $p < 0.01$; vs menopause $p < 0.0001$) assembled into thin bundles (vs reproductive age $p < 0.0001$; vs menopause $p < 0.01$) (Fig. 1D, E). These fibers became increasingly thicker with age (vs reproductive age $p < 0.01$; vs menopause $p < 0.001$). However, their assembly change upon puberty, as we observe at low magnification the densification of fibers into thickest bundles at reproductive age (vs prepubertal age $p < 0.0001$; vs menopause $p < 0.0001$; Fig. 1D).

Pore area and number analyses were used to investigate the spacing between fibers and establish the network tightness (Fig. 1F, I). This approach demonstrated that, at prepuberty, thin bundles are assembled into a tight network represented here by high numbers of pores (vs reproductive age $p < 0.0001$; vs menopause $p < 0.0001$) occupying the smallest area (vs reproductive age $p < 0.0001$; vs menopause $p < 0.001$). By comparing prepubertal and menopausal ovarian tissues, we were able to observe different pore characteristics at ×5000 and ×20,000. Indeed, while at high magnification we found that menopausal tissue is composed of a smaller pore number ($p < 0.001$) occupying a larger area than at reproductive age ($p < 0.01$), a similar observation at lower magnification revealed tighter network organization at menopause (Fig. 1F, I). This evidences a particular ECM deposition and overlapping with age, which can be translated into limited flow diffusion in menopausal tissue (Fig. 2). Such network organization could affect the elastic and viscoelastic properties of tissue, as well as limit the diffusion of water, nutrients, and cells (Fig. 2).

**Interstitial fiber orientation and straightness change with age and hormonal state.** All age groups showed directional collagen alignment, as demonstrated by fiber orientation analysis (Supplementary Fig. S1 and Table 1). They all displayed preferred average fiber orientation centered around 90°: 89.53° ± 0.33 at prepuberty; 91.63° ± 0.9 at reproductive age; and 89.14° ± 0.27 at menopause (Fig. 3). Although the difference in average fiber direction was discreet, it was statistically significant between reproductive-age and prepubertal ($p < 0.0001$) as well as menopausal ($p < 0.0001$) tissues, while no difference was detected between prepubertal and menopausal fiber orientation ($p = 0.0681$). Angular dispersion of collagen fibers was, however, unique to each age group ($p < 0.001$, all paired comparisons).

During its reproductive life, the ovary operates under cyclic tension and release synchronously with follicle development[15], which may lead to fiber straightness[16]. Our follow-up of collagen fiber waviness from prepuberty to menopause reveals increasing fiber straightness with age ($p < 0.001$, all paired comparisons), probably related to greater stretching force applied to fibers (Fig. 3C). Earlier during prepuberty, in the presence of high numbers of preantral follicles, we observed fiber rearrangement around follicles with a curvier aspect (Fig. 3). Curvy fibrils were found mainly around follicles and were analyzed separately, as discussed in the next section (Fig. 4).

**Perifollicular fiber orientation reveals directional fiber remodeling during folliculogenesis.** Analysis of collagen fiber orientation around the borders of preantral ovarian follicles at both prepuberty and reproductive age showed a dramatic reorganization of their fibrous microenvironment (perifollicular ECM[17]) with follicle stage. Indeed, the frequency of fibers oriented

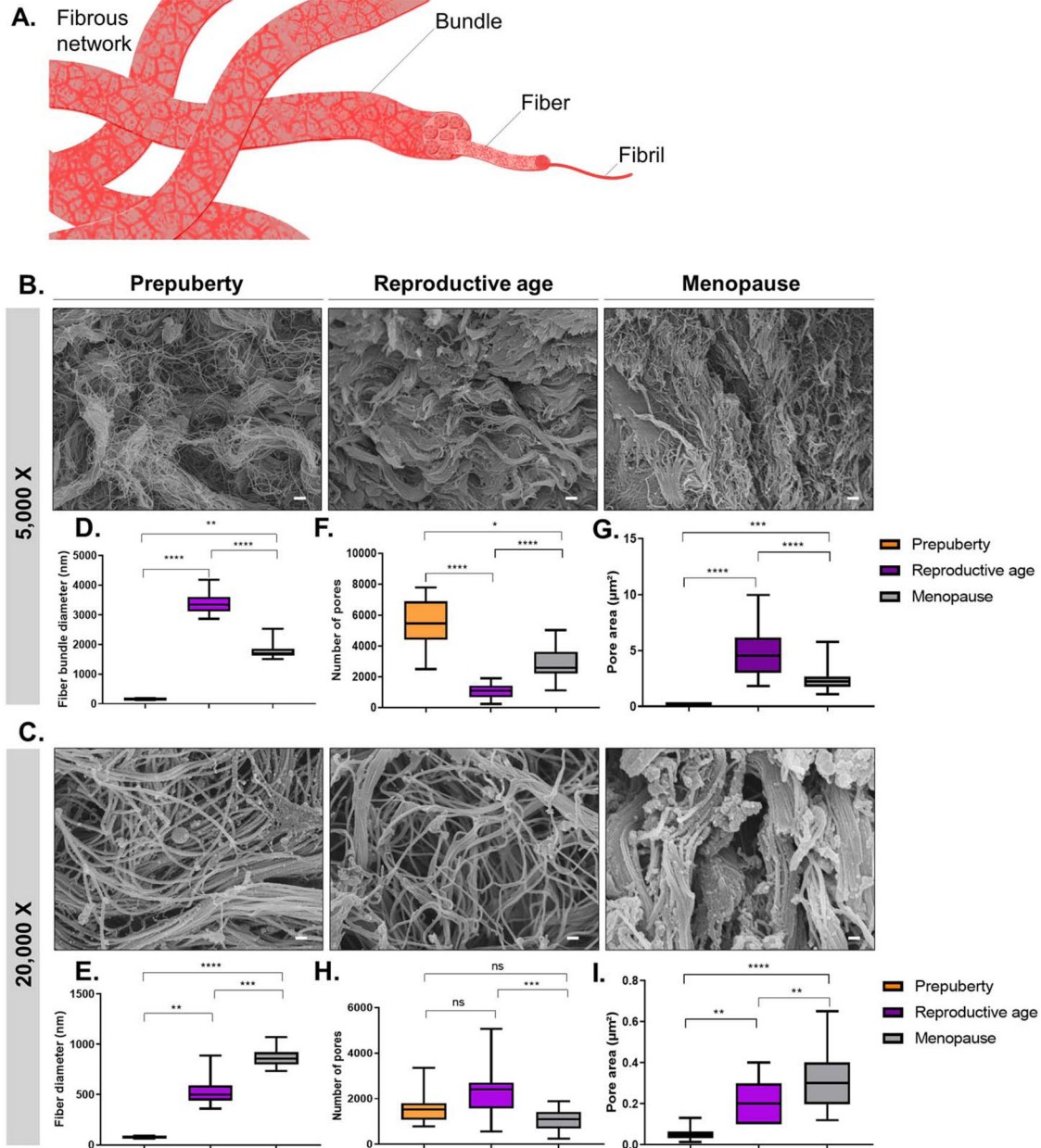

**Fig. 1 ECM microstructure and fibrous network morphology in human ovarian tissue from prepuberty to menopause. A** Schematic illustration of the fibrous network anatomy composed of fibrils, fibers, and fiber bundles, as defined in this study. SEM micrographs revealed: **B** the ECM network structure at fiber bundle scale (×5000 magnification) and **C** fiber scale (×20,000 magnification). **D, E** At prepuberty (number of biological replicates [$n$] = 5), ovarian tissue is composed of the thinnest fibers (mean ± SD: 76.1 nm ± 9.8) assembled into the thinnest bundles (160.0 nm ± 21.3), densifying upon puberty (fiber scale: 528.4 nm ± 128.0; fiber bundle scale: 3379.0 nm ± 368.8). **F–I** Fiber and bundle spacing deduced from pore number and area revealed a tight ECM network at prepuberty, illustrated here by large numbers of pores (5430 ± 1688) occupying the smallest area (0.19 μm² ± 0.05). At reproductive age ($n$ = 5), human ovarian tissue is characterized by fibers of intermediate diameter (528.4 nm ± 12.8) assembled into the thickest bundles compared to prepubertal ($n$ = 5) and menopausal ($n$ = 5) tissues, forming a looser fibrous network with greater pore spaces (4.9 μm² ± 2.4). While at high magnification, we found menopausal tissue to be composed of smaller pore numbers occupying greater areas than at reproductive age (1054 ± 470); a similar observation at lower magnification revealed tighter network organization at menopause (0.32 μm² ± 0.15). These pore-spacing differences between age groups may play a role in permeability change with age. Boxplots display 25th and 75th percentile, median (line), and the whiskers extending to the last data point not considered outlier. Fiber and pore analyses were conducted using DiameterJ. One-way ANOVA (post hoc: Tukey) was used to compare pore area. Differences in fiber morphology and pore number were analyzed by Kruskal–Wallis test (post hoc: Dunn) (*$p < 0.05$; **$p < 0.01$; ***$p < 0.001$; ****$p < 0.0001$). Scale: 1 μm.

between 0° and 90° changed significantly at the secondary stage, compared to primordial ($p ≤ 0.001$) and primary ($p < 0.01$) follicles before and after puberty (Fig. 4A–D and Table 1). Secondary follicles are characterized by a 2–3-fold larger diameter than earlier stages, which explains fiber rearrangement below 50°

consistent with follicle enlargement. When comparing prepubertal and reproductive-age perifollicular fiber directionality between follicles at the same developmental stage, we found significant differences at all stages: primordial ($p < 0.01$), primary ($p < 0.05$), and secondary ($p < 0.01$) (Fig. 4E–G). All these

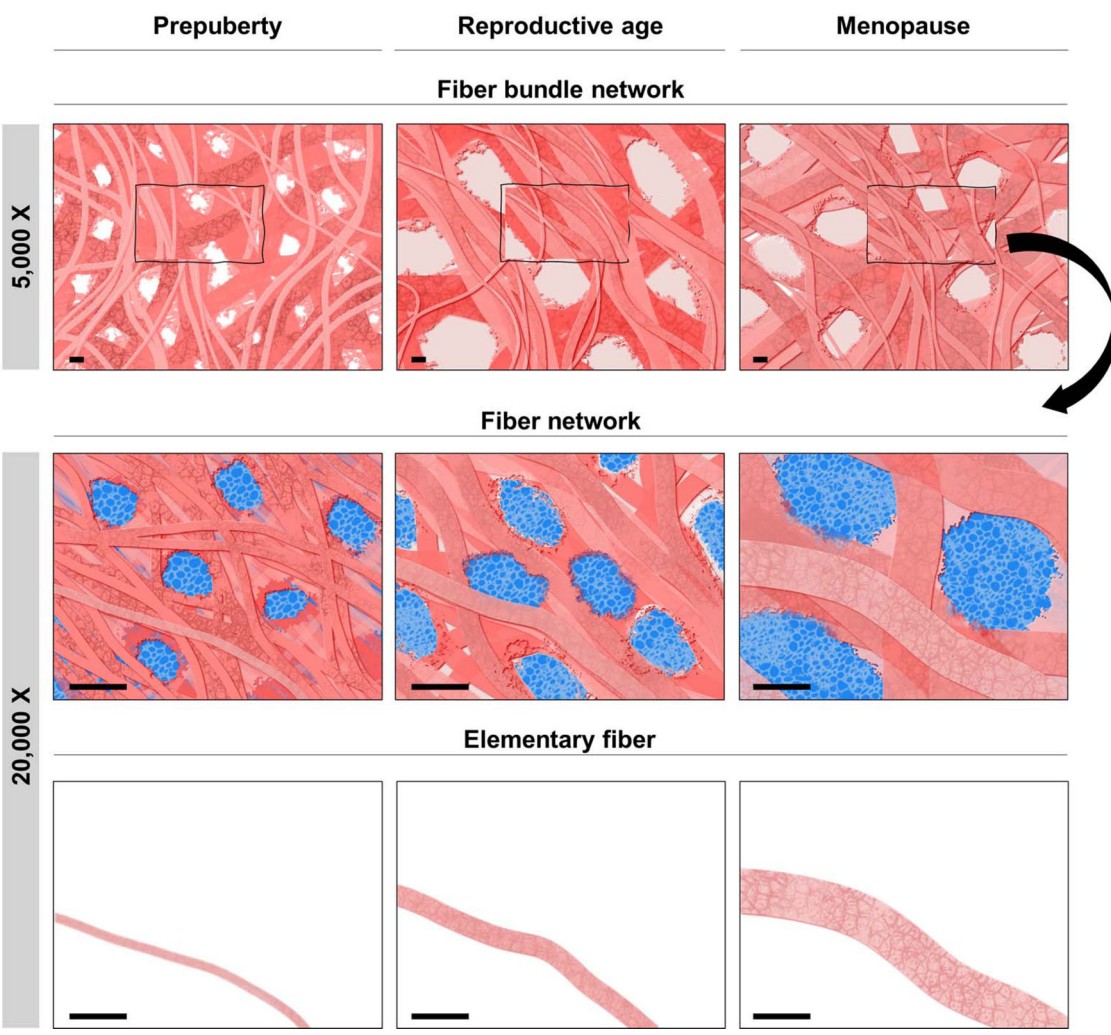

**Fig. 2 Schematic illustration of fibrous ECM structure deduced from SEM image analyses.** Simplified illustration of ECM structural remodeling with age at fiber (×20,000) and fiber bundle (×5000) scale, showing changes in fibrous network morphology (thickness, spacing). Scale: 1 μm.

**Table 1 Number of segmented and tracked fibers in perifollicular and interstitial ECM.**

| Age group | Interstitial ECM | Perifollicular ECM | | |
|---|---|---|---|---|
| | | Primordial | Primary | Secondary |
| Prepubertal | 117,005 fibers | 17,025 fibers | 10,224 fibers | 3659 fibers |
| | 74 images | 48 follicles[a] | 28 follicles | 8 follicles |
| Reproductive age | 157,445 fibers | 19,246 fibers | 10,781 fibers | 2069 fibers |
| | 104 images | 56 follicles | 31 follicles | 6 follicles |
| Menopausal | 173,325 fibers | | | |
| | 102 images | | | |

[a]One follicle per image was analyzed.

differences reveal local perifollicular remodeling of the follicle environment that depends on follicle stage and ovarian tissue age.

**Elastic and viscoelastic characteristics of human ovarian cortex**. Given that mechanical ECM properties are one of the main epigenetic environmental factors in all biological tissues, including the human ovary, respecting native ovarian mechanical features is key to engineering a functional bioinspired ovary. We therefore conducted elastic and viscoelastic measurements of human ovarian tissue and followed its evolution with age. Both of these physical characteristics coexist

and could be measured by choosing appropriate measurement parameters (Fig. 5).

We explored how the change in ECM organization is reflected in the mechanical properties of ovarian tissue (Fig. 5). Using the indentation technique, we assessed its micrometer-scale mechanical characteristics, which reflect the mechanical properties sensed by a cell in the tissue. We used atomic force microscopy (AFM) tip mounted with a round bead 5 μm in diameter. Despite complex ECM organization, the observed tissue response was similar to that of continuous media with Young's modulus (elastic constant) of around 3178 Pa (±245) for reproductive-age ovaries

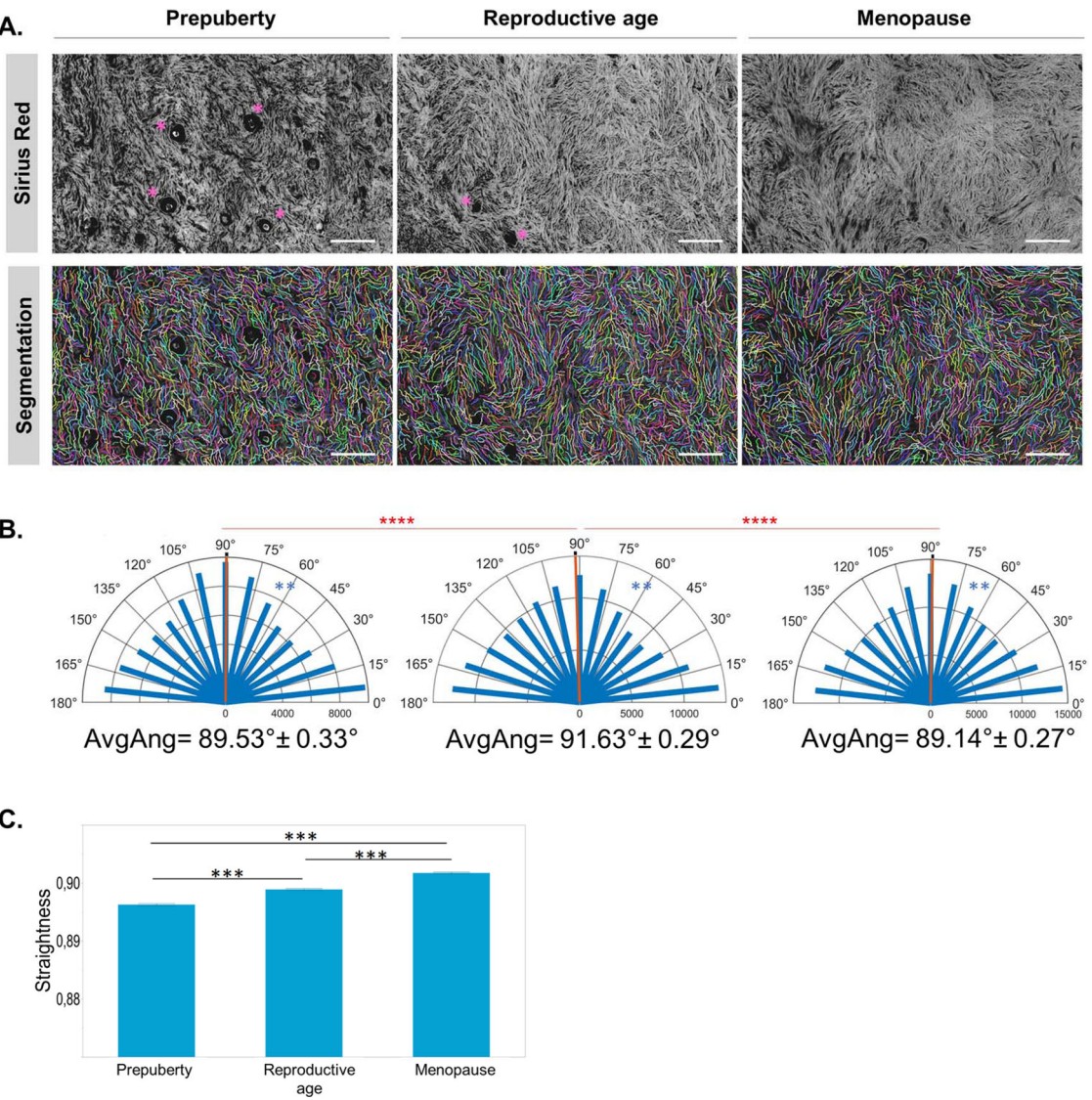

**Fig. 3 Fiber orientation and straightness in the interstitial ECM. A** Sirius Red-stained paraffin sections show collagen autofluorescence, which was used for collagen fiber segmentation by curvelet transform that enabled measurement of directional fiber distribution and fiber straightness. Asterisks point to preantral follicles. **B** Angular fiber distribution changes with age (blue) demonstrate the anisotropic nature of human ovarian tissue, but average angles change only at reproductive age (red), and no significant differences were noted between prepubertal and menopausal tissues. **C** Fiber straightness increases significantly with age ($n = 5$ biologically independent samples/group). Straightness results were extracted from segmented and tracked fibers as detailed in Table 1 and are expressed as mean ± SD. Statistical significance in fiber straightness was obtained by one-way ANOVA followed by Tukey's post hoc multiple comparisons. Three to five regions per slide were selected from Sirius Red scans for fiber orientation and straightness measurements. The Rayleigh test was applied to assess whether fibers are uniformly oriented from 0° to 180° within each age group or display a common mean direction. The Watson's parametric multiple sample test and Kuiper's test with Bonferroni correction were applied to successively compare mean directions and directional distribution between study groups (**$p < 0.01$; ***$p < 0.001$; ****$p < 0.0001$). Scale: 100 μm.

and significantly more rigid for prepubertal (6538 Pa ± 351; $p < 0.0001$) and menopausal (7117 Pa ± 714; $p < 0.0001$) ovaries (Fig. 5A–D). These changes in the mechanical properties may relate to the increased micro-porosity encountered in fertile tissues (Fig. 1G), as it was comparable in prepubertal and menopausal tissues. Other factors could also explain these mechanical changes, such as the local mechanical anisotropy as well as chemical ECM change.

We also observed a viscoelastic behavior, namely, a time-delayed response following mechanical stimulation. The viscoelastic constant was evaluated by performing stepwise deformation of the tissue. The measured viscoelastic response can be described by two relaxation time constants, T1 ~0.2 s and T2 ~2 s (Fig. 5E). At short testing period, no differences were observed between all age

groups. However, at longer time range, prepubertal tissue appears to have a significantly different viscoelastic phenotype from other age groups ($p < 0.001$), characterized by the longest relaxation time ~2.34 s against 1.95 and 1.86 s recorded, respectively, in reproductive-age and menopausal tissues (Fig. 5E).

When combining elastic and viscoelastic results, we noticed that despite significant changes in elasticity between reproductive-age and menopausal tissue, both ovaries show similar viscoelastic behaviors characterized by a short relaxation time. By contrast, prepubertal ECM is rigid, with a higher pore number, a tighter fibrous network (Fig. 2), and responds slower to deformation by regaining in a longer time its initial state (Fig. 5D, E). These two time-dependent viscoelastic responses can be explained by initial local fluid outflow from small pores

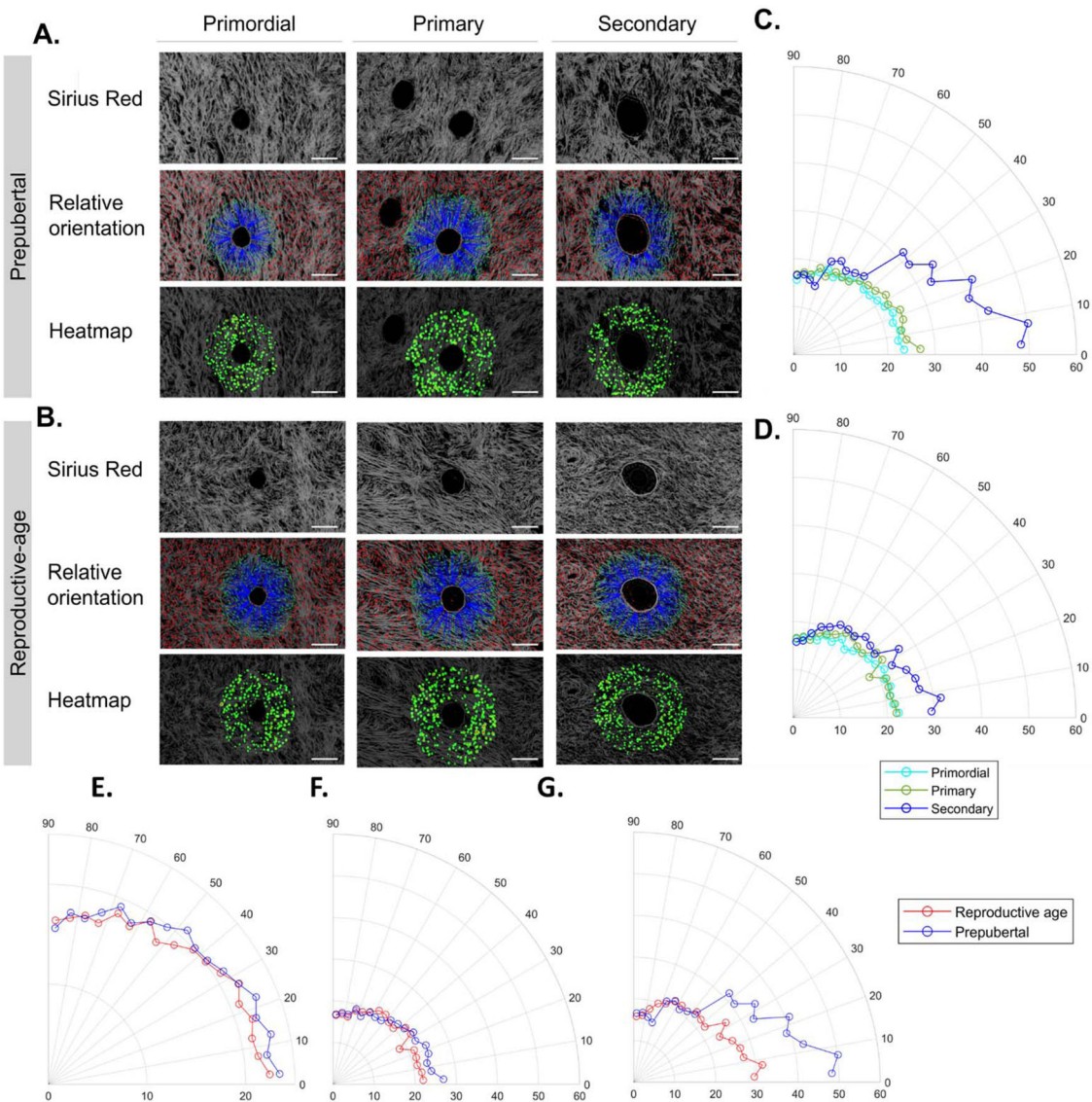

**Fig. 4 Fiber orientation in the perifollicular ECM around preantral follicles at prepuberty and reproductive age. A, B** Analysis of fiber orientation around the borders of primordial, primary, and secondary follicles at prepubertal and reproductive age using Sirius Red-stained sections. Details on the number of analyzed fibers and follicles at each stage and age are detailed in Table 1. The blue line associates the center of fibers extracted by CT-FIRE with corresponding boundary locations, while the red line shows fibers located either beyond the distance range or within the boundary; boundaries are highlighted in yellow; the heatmap uses red or warm colors to indicate larger relative angles. **C, D** Circular plots of fiber orientation frequency during folliculogenesis in prepubertal (**C**) and reproductive age (**D**) tissues revealed significant fiber reorientation at the secondary stage compared to primordial and primary stages and at both ovarian ages. One-way ANOVA (post hoc: Tukey) was used to compare fiber directionality between follicle stages within each age group. During prepuberty ($p$ values): $P_{\text{primordial-primary}} = 0.9191$; $P_{\text{primordial-secondary}} = 0.0010$; and $P_{\text{primary-secondary}} = 0.0034$. During reproductive age ($p$ values): $P_{\text{primordial-primary}} = 0.8574$; $P_{\text{primordial-secondary}} = 0.0022$; and $P_{\text{primary-secondary}} = 0.0099$. **E–G** When comparing the difference in fiber orientation around follicles from prepubertal and reproductive-age tissues at each follicular stage (**E**: primordial, **F**: primary, and **G**: secondary), we found significant differences between age groups at all stages. $P_{\text{primordial}} = 0.0026$; $P_{\text{primary}} = 0.0329$; and $P_{\text{secondary}} < 0.0001$. One-tailed $t$ test was used to compare fiber directionality around follicles at the same stage between age groups. Scale: 200 μm.

separating ECM fibers, followed by their rearrangement under stress with further fluid outflow from larger pores at the indentation site within a longer time range[16] (Fig. 2).

**Topography.** Our results reveal the roughness footprint of healthy ovary at each age using stereoscopic reconstruction of scanning electron microscopic (SEM) images (Fig. 6). We note that soft ovarian tissue at reproductive age is characterized by a smoother surface (99.16 nm ± 33.15) than rougher prepubertal (227.67 nm ± 86.99; $p < 0.0001$) and menopausal (237.80 nm ± 27.72; $p < 0.0001$) tissues, which are also more rigid (Fig. 6).

**Discussion**
In this study, we explore the role of the ECM in the functioning of human ovaries. Combining different imaging modalities, we analyze the fibrillary organization, topology, and mechanical properties of ovarian tissue before, during, and after reproductive age.

Along with the cell niche, the ECM has been shown to orchestrate cell behavior and fate[18]. For instance, the multiscale architecture of ECM collagen fibers influences cell polarity and promotes migration by providing contact guidance cues[19]. While a normal ECM can restore transformed cells to quiescence[18], a

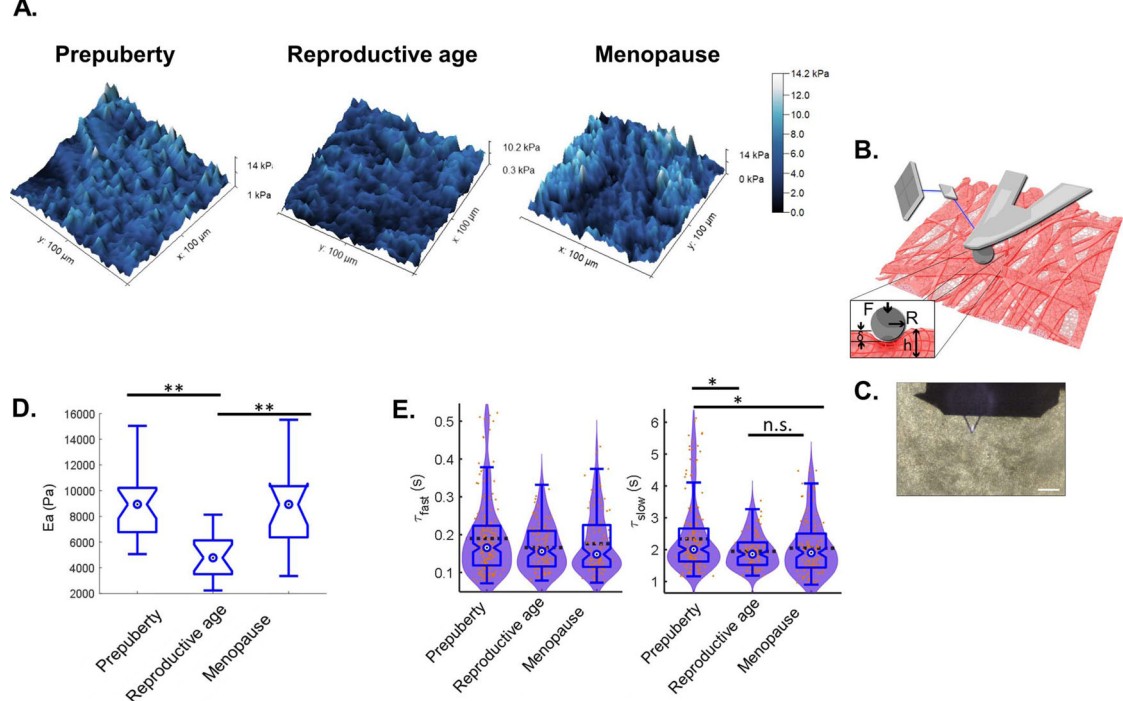

**Fig. 5 Elasticity and viscoelasticity of human ovarian tissue. A** 3D rendering of AFM maps such as surface height and color reflects Young's modulus amplitude measured using a Hertzian model. **B** Graphic representation of a Hertzian model and AFM measurement of ovarian tissue deformation under AFM spherical probe indentation: $R$ is the probe radius (2.5 μm), $F$ is the applied force, $\delta$ is the indentation, and $h$ is tissue thickness (50 μm). Deformation induces cantilever deflection. An optical system using a laser to detect the tip's deflection helped to measure Young's modulus using the Hertzian model. **C** Transmission microscopic image of the AFM cantilever positioned on top of ovarian tissue in PBS. Scale: 100 μm. **D** Elasticity measurements. $N = 5$ biologically independent samples per age group with at least three analyzed regions per sample. $P_{prepuberty-reproductive\ age} = 3.471e-5$; $P_{menopause-reproductive\ age} = 7.619e-4$. **E** Viscoelasticity measurements. Although ovarian tissue shows significant elasticity change upon menopause, no significant difference was recorded in terms of viscoelasticity ($\lambda$: relaxation time) between reproductive-age and menopausal tissues. A minimum of six measurements on at least nine different points in a sample were recorded for viscoelasticity, which was measured on the same region that was used for force maps. The graphic shows the overlay of violin plots with boxplots. Statistical significance was obtained using Kruskal–Wallis test followed by Dunn's multiple comparison correction. Boxplots display 25th and 75th percentile, median (blue circle), and the whiskers extending to the last data point not considered outlier. ns: non-significant; *$p < 0.001$; **$p < 0.0001$.

damaged ECM can trigger malignancy[20] or differentiation[21]. ECM architecture plays a role in the specialized function of the tissue of which it is a part[22]. Indeed, collagen forms thick fibers in load-bearing tendons that are aligned along the tendon to optimize force transmission and tendon strength[23]. By contrast, collagen in the cornea forms woven sheets of thin fibers that provide strength combined with optical transparency[23]. In interstitial tissue, collagen forms mostly isotropic networks, which provide mechanical strength combined with porosity to facilitate nutrient transport and cell migration.

While secretory patterns of hormones and paracrine and autocrine functions in the human ovary have been investigated for several decades, little is known about the role of ECM biomechanics and topology in ovarian function. Since its first description more than a century ago, the human ovary has been known to have a dynamic architecture, with a stiffer collagen-rich cortex and a softer medullary layer. Follicles are distributed along this collagen gradient: early preantral follicles are located in the firm cortex and late secondary and antral follicles are found in the less rigid medullary layer. Only recently, studies highlighted the fact that the mechanical properties of the ovarian ECM, including this rigidity gradient, play a crucial role in supporting follicle survival and folliculogenesis[24]. Studies are finally revealing that the ovary is a dynamic mechano-responsive organ and that crosstalk between the ECM and follicles and the ECM and ovarian cells is essential for folliculogenesis and oogenesis. Indeed, ovulation would never occur without remodeling of the dense and rigid ECM of the ovarian cortex. The "weakening" of the ECM by enzymes synthesized by cells from preovulatory follicles is vital for oocyte expulsion[25]. Another example of the influence of ECM composition and architecture on follicle development has been seen in subjects with polycystic ovary syndrome (PCOS). Ovaries from PCOS patients have a densely collagenized and thickened cortex that probably creates a biomechanically non-permissive environment, possibly altering mechanical signaling. This environment is likely to play a role in boosting numbers of growing follicles through a cascade of events, culminating in increased secretion of growth factors, which leads to primordial follicle activation and development[26]. PCOS ovaries are also characterized by anovulation, probably a result of the lack of degradation of the fibrous ECM[11] or an undiagnosed abnormal ECM architecture.

Kawamura et al.[27] reported an increase in primordial follicle activation in PCOS patients after cutting ovarian tissue into small cubes. This procedure was found to modify ovarian mechanical forces by releasing tension on ovarian cells and disrupting the Hippo signaling pathway, characterized by polymerization of globular actin to filamentous actin. Actin polymerization may connect biophysical changes with suppression of Hippo signaling, increasing nuclear YAP concentrations, and stimulating follicle growth[28]. Moreover, Telfer and McLaughlin (patent WO2014043835A1) proved that stretching small strips of

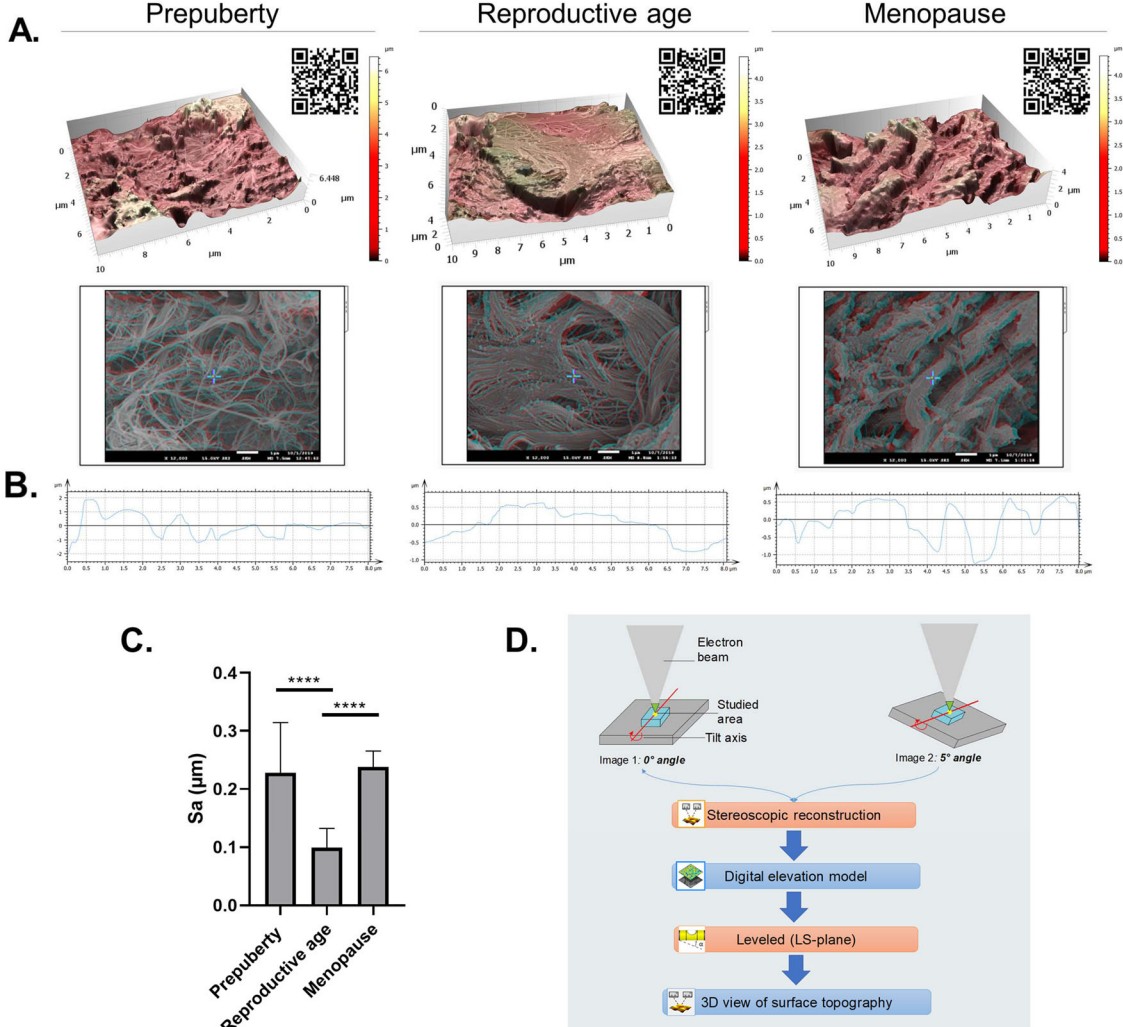

**Fig. 6 Topography and surface analyses. A**, **D** 3D rendering of surface topography from stereoscopic reconstruction of two SEM images acquired using eucentric rotation around the *y* axis with a 5° tilt at ×12,000 magnification. Color scales indicate the height range on each surface. Scale on SEM micrographs corresponds to 1 μm. QR codes open video animation of 3D topography representations. **B** Extracted rugosity profiles from 3D reconstructions. **C** Surface roughness was measured by calculating the arithmetic average of 3D roughness (Sa) according to calculation conditions defined in ISO25178. At least three regions were acquired per sample (*n* = 5 biologically independent samples/age group). **D** 3D stereoscopic reconstruction pipeline. One-way ANOVA (post hoc: Tukey) was used to compare surface roughness (****$p < 0.0001$).

cortical ovarian tissue in one direction by >10% of the initial length prior to in vitro culture enhances primordial follicle activation to the secondary stage. These initial observations highlight the role of ECM architecture as a regulator of cellular/follicle behavior in the ovary, demonstrating the importance of microenvironmental cues to female fertility. Although mechanical stimulation is mediated by specific architecture in different tissues[29], there is still a lack of understanding of ovarian ECM topology that needs to be addressed. We therefore believe that identifying differences in the ECM architecture of ovaries before and after puberty and during menopause will help us elucidate how biochemical regulation can be correlated to ECM morphology. Our spatiotemporal follow-up of fibrous ECM morphology enables phenotyping ovarian tissue at different ages, shedding light on the possible effect of the fibrillar structure on the signaling of growth factors binding to the fiber surface. Here the ECM brings active molecules into close proximity with the cell surface and facilitates interactions between growth factors and integrins or sequesters them within the fibrous network and tight pores[30].

Our results revealed a unique ECM architecture at reproductive age, where fibers of intermediate diameter are assembled into thickest bundles compared to prepubertal and menopausal tissues. This is probably due to the estrogen surge at reproductive age that stimulates the activity of LOX[31], the crosslinker of collagen. These data are consistent with previous collagen crosslinking analyses under polarized light[17]. Fiber and bundle spacing deduced from pore number and area show a looser fiber network at reproductive age compared to other age groups. While fiber thickness reflects fibril crosslinking and participates in the physical properties of tissue, multi-scale pore size and distribution contribute to diffusion and access of key molecules (nutrients, waste, hormones, and oxygen) to/from cells. Hence, changes in tissue pore spacing from prepuberty to menopause can be translated into changes in permeability and molecule selectivity from one age to another. This information may be useful for the pharmacokinetics of future ovarian treatments that might require age-personalization.

Pore geometry is also dependent on fiber orientation. Our results demonstrated the anisotropy of human ovarian tissue

characterized by directional variations in tissue organization throughout female life, with a specific arrangement at each age. The interwoven anisotropic fiber structure of the human ovary provides the tissue with great adaptability to multidirectional loads[32]. Interestingly, reproductive-age tissue fibers follow a common average orientation, which is significantly different from prepubertal and menopausal age groups that share comparable average fiber orientation. Taking into account the implication of pulling forces by cells in fiber orientation[33,34], this diversity in directional fiber distribution reveals a different multidirectional mechanical load in each group. This further delineates the peculiar organization of the ECM related to ovarian activity at reproductive age and hormonal homeostasis and highlights possible differences in force transmission between age groups[29]. Considering the uniqueness of fiber orientation in ovarian cortex at reproductive age, fiber directionality may become a hallmark of female fertility, as is the case for breast cancer prognosis[35,36]. Although detected differences in average fiber angles are in the order of 2°, no available data in the literature rule out the relevance of this discrete difference in the physiological function of ovarian tissue. Future studies need to elucidate the involvement of fiber orientation and stability range in functional tissue.

By comparing local fiber directionality around follicle borders at primordial, primary, and secondary stages, we noted that, before and after puberty, secondary follicles appear to modify their microenvironment arrangement locally compared to follicles at earlier stages of development, by reorienting the majority of collagen fibers below 50°. Our results complement and corroborate in vitro studies demonstrating that regulation of ovarian follicle development depends on the architecture of the perifollicular ECM, in addition to endocrine- and paracrine-acting hormones[37]. Beyond providing structural support for follicle formation and growth, the fibrous ECM network acts as a reservoir for paracrine and endocrine signals inside the ovary and permits or restricts their access to cells within follicles. Similarly to other organs, the ovarian ECM is prone to mechanical and enzymatic remodeling in different physiological and pathological conditions[37]. Hence, our observation of differences between perifollicular ECM architecture in prepubertal and reproductive-age follicles at similar stages could be a cause of differences in molecular signaling and interfollicular communication, responsible for the life cycle of preantral follicles. Throughout folliculogenesis, ECM architecture ensures appropriate hormone secretion, somatic cell differentiation, and oocyte maturation[11]. In conjunction with this activity, the ovary accommodates a large reserve of inactive primordial follicles that contain nongrowing oocytes and nondividing, flattened (squamous) pregranulosa cells surrounded by a basal lamina. Following entry into the growing follicle pool upon activation, squamous pregranulosa cells surrounding the oocyte become cuboidal granulosa cells (primary follicles). They proliferate to form multiple layers, a morphological hallmark of secondary follicles. At this stage, the theca cell layer is recruited from the stroma to surround the basal lamina, which involves intricate polarity changes and cell migration processes. Fortunately, advances in in vitro follicle culture have allowed us to decipher the role of ECM architecture in follicle and ovarian development. It was found that cell adhesion, supported by the ECM, induces changes in cell shape and motility necessary for various cellular functions during folliculogenesis. A number of studies have also shown that in vitro culture of granulosa cells plated on matrices of ECM proteins of various densities also leads to alterations in cell adhesion and shape[38], highlighting the importance of using biomimetic scaffolds to achieve completion of folliculogenesis in vitro. This can only be accomplished by following our blueprint of fiber orientation and angular density of the perifollicular ECM and our previous description of elastic

matrisome spatiotemporal changes[17]. Our results also revealed that, at all stages of folliculogenesis, prepubertal follicles are surrounded by a higher density of fibers. This can predict a non-permissive environment at a very early stage of preantral development, as already suggested[39], or peculiar architectural regulation of molecular signaling and transmission activity, as described above, which both play key roles in follicle quiescence and ovarian reserve preservation during prepuberty. Moreover, follicle survival has been described in vitro as dependent on scaffold geometry, where greater adhesion of follicles to ECM fibers oriented below 90° resulted in the highest survival rates[40]. By comparing local fiber directionality around follicle borders at primordial, primary, and secondary stages, we noted that, before and after puberty, secondary follicles appear to modify their microenvironment arrangement locally compared to follicles at earlier stages of development by reorienting the majority of collagen fibers below 50°. This could indicate that follicles at this stage require a higher degree of fiber contact and adhesion signaling[36] to thrive and complete their development and maturation toward ovulation[41]. Indeed, this was previously suggested by Laronda et al. when designing a three-dimensional (3D) bioprosthetic ovary made of 3D-printed gelatin fibers encapsulating preantral mouse follicles, where fibers oriented at 30° and 60° offered higher follicle-scaffold contact than at 90° and resulted in offspring[40]. Comparably, our results offer an opportunity to fine-tune 3D fiber printing and architectural design in accordance with native fiber and pore geometry of human ovarian tissue.

ECM fibers possess microscale and nanoscale features that provide cells with topographic stimuli and act as instructive cues to influence gene expression, cell behavior, and morphology[42]. Topographical deregulations have been described in several pathological conditions, which underlines the importance of topographical ECM homeostasis, as evidenced by emerging publications on this topic[43–45]. The mechanism of topographical cell sensing on a nanoscale level is believed to be due to the clustering of integrins and other cell adhesion molecules, because surface roughness ensures particular spacing of adhesive sites[16]. Growing evidence points to the involvement of topography in cell morphology and Hippo pathway signaling[46]. Mechanical approaches are now clinically used to disrupt ovarian Hippo signaling and promote follicle growth as infertility treatments, by releasing mechanical tension applied to quiescent follicles[47]. Indeed, maintenance of follicle quiescence in infertile, prepubertal, and menopausal subjects was thought to be due to mechanical retention by the ovarian ECM[10]. Although this theory has been supported by many researchers over the past decade, our knowledge of healthy ovarian tissue biophysics remains limited. The only rheological study conducted to date used multi-modal magnetic resonance elastography on ovaries of women diagnosed with PCOS vs age-matched controls[48]. The results revealed an overall higher degree of stiffness in PCOS tissues. Despite the non-invasiveness of this technique and clinical transferability, it provides only bulk rheology. Depth penetration of acoustic waves was patient specific in some cases, yielding low mathematical confidence, as reported by the authors[48]. Nevertheless, this work shed light on ovarian rigidity as a clinical sign of spreading ovarian pathology.

In this study, we chose to use AFM for its high sensitivity and effectiveness in distinguishing small mechanical changes between and within tissues at micro-scale resolution and hence adequately assessing the mechanical stimuli felt by cells[49]. Elasticity measurement evidenced a relatively rigid ovarian tissue at prepuberty, softening significantly at reproductive age, then stiffening considerably upon menopause. Again, ovarian tissue was found to display its own unique mechanical phenotype during

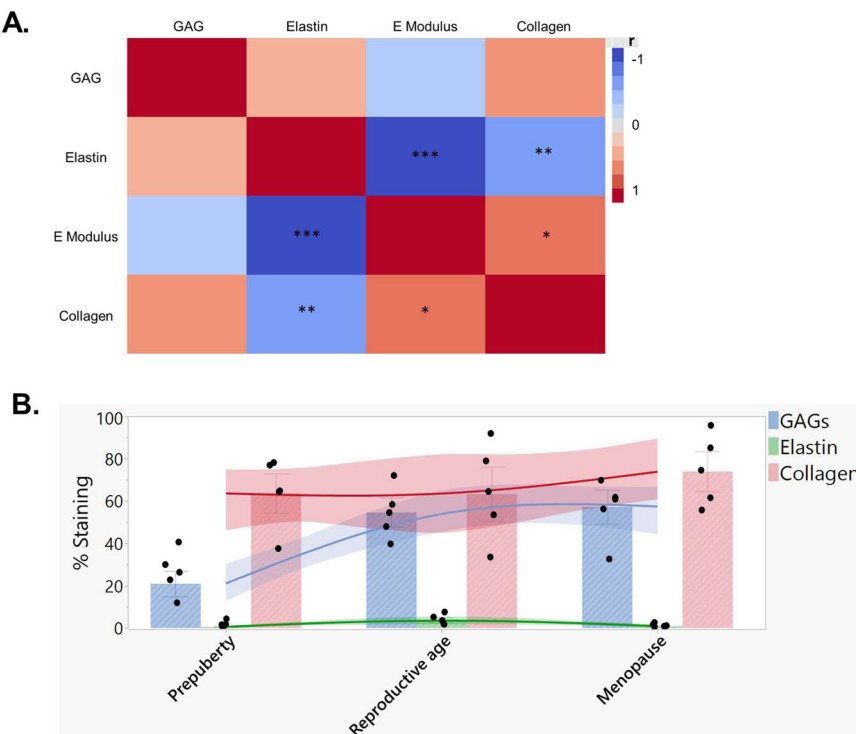

**Fig. 7 Linking biochemical to biophysical properties. A** Correlation matrix of elastic matrisome components and apparent elastic modulus measured in study subjects. Histological staining was used to characterize glycosaminoglycans (GAGs), collagen abundance, and immunofluorescent staining of elastin[17]. Computer-assisted quantification of staining was based on the following equation: % stained area = (stained area/region of interest (ROI) area) × 100 expressed as a percentage. A correlation matrix was constructed using JMP Pro 14.3.0 (SAS, France, Grégy-sur-Yerres). Spearman's correlation was used to measure the strength and direction of association ($r$) between each pair of variables. $P_{collagen-elastin} = 0.0058$; $P_{Emodulus-collagen} = 0.0148$; $P_{Emodulus-elastin} < 0.0001$. **B** Changes in collagen, elastin, and GAG levels with age are summarized in bar charts, presented as mean ± SD, $n = 5$ biologically independent samples per age group. The data plot fits a smoothing spline with a lambda value of 0.05. The curves also include the bootstrap confidence region for each fit generated by JMP Pro 14.3.0. *$p < 0.05$; **$p < 0.01$; ***$p < 0.0001$.

reproductive age. These differences are not only structure dependent but also related to biochemical differences in ECM composition, as previously demonstrated in our follow-up of variations in elastic matrisome components from prepuberty to menopause[17], and further corroborated by our samples (Fig. 7 and Supplementary Fig. S2). While collagen levels that are associated with tensile strength[50] rose by <5% after puberty, we observed a very significant difference in glycosaminoglycans (GAGs) in reproductive-age tissues. The increase was around 30% higher than prepubertal tissue, which explains the elasticity differences between groups[17] (Fig. 7). Thanks to their water sequestration properties, through their highly negative charges, GAGs generate a water-rich network with unique elastic properties, as seen in reproductive-age tissue[51]. Conversely, the levels of collagen and GAGs are maintained from reproductive age to menopause. Thus, a plausible explanation for the elasticity change accompanying aging is the significant drop in elastin upon menopause[17] (Fig. 7).

Elastin fibers provide tissues with recoil to allow repeated stretching and contraction, and their aging-associated degradation appears to have profound effects on ovarian tissue elasticity. Moreover, ovarian rigidity seems to act in synergy with surface roughness. Our results disclose surface characteristics unique to reproductive-age ovaries characterized by a smoother surface compared to other age groups.

Despite widespread application of AFM for mechanical measurement of biological tissues, the majority of existing studies are limited to the framework of standard Hertzian contact mechanics, measuring only the apparent elastic modulus of materials and disregarding their viscoelastic properties. Here, in addition to the

apparent elastic modulus measurement, we also investigated the viscoelasticity of human ovarian tissue. ECM viscoelasticity depends on fluid retention and flow through the porous matrix, as well as rearrangement of individual ECM fibers to absorb stress[52]. During short deformation period, the similar viscoelastic behavior observed between age groups can be explained by fast rearrangement of the flexible chains of ECM macromolecules[16]. At longer time range, disentanglement of the chains takes place, and relaxation time associated with this process shows a stronger dependence on the ECM architecture rather than ECM biochemistry[53]. At this time point, the viscoelasticity of reproductive-age and menopausal tissues is comparable, although elasticity changes significantly upon menopause. This finding further underscores the relevance of recording both measurements. Conversely, prepubertal tissue has a slower viscoelastic response that we can explain by its higher fluid sequestration compared to the other groups. This is related to its unique tight fiber network and small pore size, promoting long-term water retention and extending relaxation time.

The described age-dependent characteristics of ovarian ECM impact differently the follicle ovarian reserve and consequently affect the ovarian activity. While reproductive activity ceases upon menopause, the follicle reserve is not always completely consumed and some preantral follicles remain quiescent for the rest of a woman's life. We speculate that menopausal ovarian tissue can be subject to formation of advanced glycation end-products (AGEs), which are characteristic of aging ECM[54]. These AGEs are able to crosslink and thereby stiffen collagen structures, consequently hindering their movement, as observed in the menopausal group. Certainly, such a stiff matrix incompliant to

deformation cannot accommodate the developing ovarian follicles with their evolving size.

Our study might therefore be the first step to unlocking the mysteries of follicle quiescence and envisaging an extended reproductive lifetime, and even ovarian rejuvenation, by paving the way for possible therapeutic targets at the ECM level to treat infertility. The shared information herein will serve to develop new strategies for infertility prognoses based on ECM biophysics, as was done for early breast cancer, based on tissue rigidity, fiber orientation, and straightness[35,55,56].

In conclusion, our study provides conclusive proof of a link between ECM rigidity and fertility by comparing different stages of ovarian transformation related to a woman's reproductive life. A clearer understanding of the mechanisms by which ovarian cells respond to ECM cues should also bolster efforts to judiciously prioritize material properties for therapeutic benefit. Fortunately, new techniques are increasingly being developed to create scaffolds with precise fiber morphology[30,57], network architecture[58], topography[59], elasticity, and even viscoelasticity[60]. Our work emphasizes the importance of nanometer-scale fibril orientation and ECM porosity. Indeed, development of artificial ovarian tissue will require technology capable of highly precise scaffold printing, such as nano 3D printers and FluidFM. We are confident that our work represents the missing piece of the puzzle to leap to the next generation of bioinspired tissue-engineered ovaries.

## Methods

**Ovarian biopsies included in the study**. Use of human ovarian cortex was approved by the Institutional Review Board of the Université Catholique de Louvain on May 13, 2019 (IRB reference 2012/23MAR/125, registration number B403201213872). Ovarian tissue from prepubertal girls was donated by patients treated at the Children's Hospital and the Department of Obstetrics and Gynecology of the University Central Hospital of Helsinki (Finland). Pediatric patients from Children's Hospital in Helsinki participated in a fertility preservation program and a research project approved by the Ethics Committee of Helsinki University Central Hospital (license number 340/13/03/03/2015). Written informed consents for pediatric patients were given by their guardians and by all age-appropriate patients.

Ovarian biopsies were taken from prepubertal (mean age [±SD] = 7 ± 3 years), reproductive age (mean age [±SD] = 27 ± 5) and menopausal (mean age [±SD] = 61 ± 6 years) patients after obtaining their informed consent. All participating adult subjects were undergoing laparoscopic surgery for benign gynecological diseases not affecting the ovaries. Prepubertal tissue was derived from young cancer patients scheduled for ovarian cortex cryopreservation as a fertility preservation strategy, before being subjected to acute gonadotoxic cancer treatments.

All samples were cryopreserved by slow freezing[61] and kept frozen until the day of their analysis. Tissues provided from the same patients (n = 5 per age group) were investigated by SEM (fiber, pore, and topography analyses) and AFM. A larger number of paraffin-fixed biopsies (prepubertal, n = 16, reproductive age, n = 21, and menopausal, n = 24) obtained from the biobank of St-Luc's Hospital were used to complete our fiber orientation evaluation, as described below.

Patients enrolled in this study were carefully selected to avoid any possible bias or heterogeneity related to non-synchronous cycle phases between reproductive-age patients or use of hormone replacement therapy after menopause. Therefore, only fertile patients under ovarian contraceptive treatment and menopausal patients not taking hormonal replacement therapy were included in this study.

**Fiber and pore characterization**. SEM images were analyzed using the Diameter J plugin[62] of ImageJ software V1.48 (Wayne Rasband, NIH, USA) to measure fiber thickness as well as pore space and number at ×5000 and ×20,000 magnification. Briefly, all images were converted to binary images (black and white pixels only) and segmented into background and foreground, according to 1 of the 24 segmentation methods generated by Diameter J[62]. Segmented pictures contain only black and white pixels, with black pixels representing the background and white pixels representing fibers. After visual inspection of all generated binary pictures, only those that closely resembled to the original SEM image were manually selected for further analyses in batch mode.

Fiber thickness (diameter) was calculated using super pixel determination[62], and mean fiber diameter was calculated by fitting a Gaussian Curve to the radius data and finding the curve's mean value. Pores were considered as black pixel clusters. The algorithm counts the number of pixels in each cluster not touching

the picture frame edges and then reports their area, which equates to pore area. Mean pore area was calculated by averaging all cluster areas.

**Scanning electron microscopy**. Ovarian samples were fixed in Karnovsky solution immediately upon thawing[61] overnight at 4 °C. After fixation, the ovarian fragments were washed five times for 5 min each, before being immersed overnight in a solution of 30% glycerol in 0.1 M phosphate-buffered saline (PBS) at 4 °C, then plunged into liquid nitrogen and cryofractured to examine the cortex beyond the epithelial surface. The samples were again washed in fresh 0.1 M PBS three times for 5 min each, before post-fixation in a solution of 2% osmium tetroxide (OsO4) in 0.1 M PBS in the dark to preserve their structure and increase conductivity. After 1 h, the washing step was repeated three times, as previously described. The fragments were then dehydrated using a graded series of ethanol (30–100%), dried to the critical point, coated with gold film (Quorum Q150T ES, Quorum Technologies, East Sussex, UK), and scanned by SEM (JSM-7500F, JEOL, Tokyo, Japan) with electron beam energy set at 15 keV. At least three regions were acquired from each sample under SEM at 3 different magnifications: ×5000, ×12,000, and ×20,000.

**Immunofluorescence staining**. After deparaffinization of paraffin sections in Histosafe (Yvsolab SA, Beerse, Belgium) and ethanol, autofluorescence was quenched by 10-min incubation in 50 mM NH4Cl, and endogenous peroxidase inhibition was achieved after 30 min in 0.3% H2O2. Sections were then subjected to water bath antigen retrieval in 10 mM citrate buffer (pH 5.7) containing 0.1% Triton or Tris-EDTA buffer (pH9.0), before blocking of nonspecific antigen-binding sites (Tris-buffered saline (TBS) solution containing 10% normal goat serum and 1% bovine serum albumin (BSA)). Monoclonal anti-elastin (dilution 1:250, Abcam, EPR 20603) was incubated overnight at 4 °C in a humid chamber in TBS containing 1% BSA and 0.1% Tween-20 and detected by EnVision rabbit horseradish peroxidase (HRP)-conjugated secondary antibody (Dako, K4003, dilution 1:1, Glostrup, Denmark) for 1 h at room temperature. HRP was then visualized by tyramide signal amplification using Alexa Fluor-conjugated tyramide reagent (Invitrogen, dilution 1:1, AF488 B40953). Finally, nuclei were counterstained with Hoechst 33342 (dilution 1:1000, Thermo Fisher Scientific, Waltham, MA, USA) and mounted with HIGHDEF IHC fluoromount (Enzo, New York, USA). Negative controls were conducted by adding rabbit IgG (Dako, Carpinteria, USA) instead of the primary antibody, while human testis was used for positive control. Stained slides were digitized by automated whole-slide image capture with a Pannoramic 250 Flash III scanner (3DHISTECH Ltd, Budapest, Hungary).

**Sirius Red staining**. Five-μm-thick paraffin sections were deparaffinized, rehydrated, and treated with a saturated picric acid solution containing 1.3% Sirius Red for 1 h at room temperature for collagen staining. Mounted slides were digitized at ×20 magnification by automated whole-slide image capture using a Mirax digital slide scanner (3DHISTECH Ltd, Budapest, Hungary). Images were acquired at the following settings: excitation length, 450 nm; emission length, 550 nm; and exposure time, 100 ms.

**Orientation analysis**. Three to five regions per slide were selected from Sirius Red scans for fiber orientation analysis, using CaseViewer V2.4 (3DHISTECH Ltd, Budapest, Hungary). CT-FIRE V2.0 Beta[56] Matlab-based open source software was used for fiber tracking, segmentation, and orientation analysis. The software combines curvelet transform (CT), a fast discrete preprocessing mechanism that de-noises the image and enhances fiber edges, and fiber tracking (FIRE), an algorithm to extract single fibers from fiber networks and measure individual fiber metrics. The fiber angle in CT-FIRE is defined as the angle between the line linking two endpoints of a fiber and the positive horizontal direction or positive X axis (Supplementary Fig. S1 and Table 1). To calculate average fiber orientation and its confidence interval, and visualize data on semi-polar plots, we used the Matlab toolbox CircHist[63].

To investigate fiber organization and remodeling around primordial, primary, and secondary follicles at prepuberty and reproductive age, follicles were selected from ×40 magnified images and their borders were delineated manually and selected as regions of interest (ROIs). CurvAlign V4.0 Beta[56] was applied to study fiber orientation relative to ROI borders (Table 1).

To visualize data on fiber orientation against follicle borders, we used the Polarplot function on Matlab 2019a (MathWork, Eindhoven, the Netherlands).

**Fiber straightness analysis**. Fiber straightness is defined as the ratio of fiber Feret length divided by its geodesic length, and its value ranges from 0 to 1 (Table 1). A perfectly straight fiber would have a straightness value of one and increasing complexity (non-straightness) as the value approaches zero. This parameter was quantified using the CT-FIRE software on Sirius Red-stained slides.

**Sample preparation for AFM**. After thawing, cryopreserved ovarian biopsies were immediately embedded in optimal cutting temperature compound (OCT Tissue-Tek, VWR International, Edmonton, Canada) and snap-frozen in liquid nitrogen, without prior labeling, fixation, or dehydration. Fifty-μm-thick ovarian cortex

cryosections were obtained using a cryostat (Microm HM 560, Walldorf, Germany) and mounted on Superfrost Plus glass slides (Menzel-Glaser, Germany). The first 200 μm from the surface was cut away to go beyond the mesothelial layer, and tissue slides 300–500 μm in depth within the ovarian cortex were selected for AFM analyses. On the day of AFM analyses, tissue sections were incubated at room temperature for 5 min to thaw the OCT compound and reinforce tissue adhesion to slides. The slides were then rinsed in pure water several times to remove excess OCT and incubated in PBS at room temperature for 20 min to equilibrate ion charges while conducting AFM calibration steps, as detailed below. Finally, the samples were mounted for AFM and covered with fresh PBS to conduct biomechanical characterization.

**AFM measurements**. The 50-μm-thick cryosections of ovarian tissue were kept at −80 °C until the start of the experiment and prepared as described in the section "Sample preparation for AFM." Rheological properties were measured with the NanoWizard 1 BioScience (JPK Instruments) AFM operating in force spectroscopy mapping mode, as previously described before[64–66]. Force–indentation curves were collected using a rectangular silicon nitride cantilever with a 0.07 N/m nominal spring constant, and 5-μm-diameter borosilicate glass spherical particle attached to the tip. A spherical probe was selected because it produces considerable force with minimal damage to the surface, so is ideal for compliant materials like ovarian tissue[67]. The measured spring constant was very close to the nominal value of 0.07 ± 0.005 N/m using the thermal tune method. This step is designed to transform the laser position signal on the AFM receptor into deformation of the cantilever by evaluating its spring constant in order to translate it into a force. The same cantilever was used for all experiments (Novascan, Ames, IA, USA). The indentation force was set to limit maximum indentation to ~0.5–1 μm, so the bead–tissue contact area during all tests never exceeded 7.068 μm². The experiments were performed at room temperature using 1× PBS buffer. To prevent mechanical modifications arising from tissue damage, the tests were limited to 40 min per sample. This included at least three $100 × 100$ μm² force maps and 9 positions each with 6 repetitions of viscoelasticity measurements. No significant difference was observed between measured areas within the same tissue sample, indicating that the ovarian tissue was homogeneously mechanically averaged over the spatial window of $100 × 100$ μm². Further mapping would have led to experimentally induced variability. For a $100 × 100$ μm² area, we performed $50 × 50$ measurements, resulting in 2500 force–indentation experiments. The tissue sections were immobilized on Superfrost Plus glass slides Adhesion slides (Menzel-Glaser, Germany). Tight interaction between the samples and the glass did not induce local ECM modifications (artificial solvents, tapes, and glues could dissolve or otherwise disturb the physiological state of the ECM, leading to local softening affecting AFM measurements).

Prior to tissue measurements, AFM sensitivity and alignment were ensured with an undeformable (compared to the biological sample) empty glass slide in PBS. The difference between cantilever deflection on a rigid surface and the compliant tissue sample illustrates the deformation of the tissue under the bead load. The force–indentation (F–δ) curves can be fitted with a single exponential following the Hertzian contact model. The Hertzian contact model generates the relationship between the applied force, F, and the resulting indentation, δ, allowing the extraction of an apparent Young's modulus, Ea (a for apparent), namely, the correlation constant between the force and area of indentation. Ea is a standard measure for soft tissue elastic properties. We considered the tissue to be incompressible (assumed Poisson ratio: 0.5). The Ea of the probed samples was calculated by fitting the contact part of the measured approach force curves to a standard Hertzian model for a spherical indenter (tip) of radius, R[68]:

$$F = (4Ea\sqrt{R})*(\delta^3/2)/(3(1-v^2)) \qquad (1)$$

$$Ea = 3(1-v^2)F/(4\sqrt{R})*(\delta^3/2) \qquad (2)$$

where $v = 0.5$ is the Poisson ratio and δ is the indentation depth, calculated by subtracting cantilever deflection from tip displacement. The Hertzian model is typically used to determine contact between two linear elastic bodies. As such, several assumptions had to be checked concerning our biological tissues: (i) that they displayed linear elasticity at the scale examined and (ii) that the nonhomogeneity of ovarian tissue (as a composite material) was negligible at the scale examined. Our data are well fitted by the Hertzian model, which is the best approximation for Young's modulus of biological tissues[69].

Young's modulus is presented using a violin distribution plot overlaid with mean and median values calculated over F–δ curves (i.e., pixelwise). For topographical reconstructions, the height of each point was determined by the point of contact from the F–δ curve, with each contact point issuing from the same curve used to determine Ea. Stiffness data were projected onto topographical maps using Matlab.

To measure the viscoelasticity of ovarian tissue, we conducted repeated and successive long indentation cycles, followed by partial force release. For indentation portion, force was kept constant, while for the release portion, deformation was constant, allowing us to monitor the evolution of both deformation and force. The viscoelastic measurement (force spectroscopy) immediately followed elastic measurement (force mapping scan) and was performed in two points of the same

tested region according to a force ramp design loop of ~240 s duration: cantilever extension with force increase (4-μm indentation, 100 ms); constant height maintenance at the contact point (20 s); cantilever retraction and force decrease (0.8 μm, 20 ms); and finally constant force maintenance (20 s). Data sampling frequency was set to 4000 Hz.

Viscoelastic materials retain their shape after deformation, but with a time delay, as shown by the relaxation constant. In slow deformations, we can assume that the ECM is incompressible (Young modulus measurements). However, fast viscoelasticity measurements demonstrate that the tissue may be (reversibly) compressible in short time scales. In this context, viscoelasticity was described by the generalized Maxwell model composed of spring constants (elasticity) and dashpots (relaxation time). Relaxation time was obtained by fitting the modified Kelvin–Voigt model (spring and dashpot connected in parallel), assuming exponentially decaying force at a constant deformation[64]. Thus, it permits establishing bulk elastic constant and relaxation time. The modified Kelvin–Voigt model can be described by the following:

$$F(t) = a\Delta x(1-e^{-bt}), \text{ where } b = \frac{K_1+k_2}{n} \text{ and } a = \frac{K_1 k_2}{K_1+k_2} \qquad (3)$$

We focused on the part of a curve where the deformation (Δx) was kept constant and the force evolved as a negative exponential (Supplementary Fig. S3). We observed that our experimental data are best fitted by two different relaxation times, which could be described by the generalized Kelvin–Voigt model. The double exponential fit modeled our data better than a single exponential[64], suggesting the presence of at least two processes behind the viscoelastic response. For more details on the model fitting, please consult the Matlab scripts at https://github.com/inatamara/AFManalysisMatlab [70].

**Topography measurement**. Stereo pairs of SEM images of the same tissue region at ×12,000 magnification were created by tilting samples by +5° along the X axis (Fig. 7D). By rotating the stage between two image takes, we ensured that a significant feature of the sample was always positioned at the same point and that contrast and intensity between images were similar in order to facilitate point matching and guarantee eucentricity. Mountains SEM software 8 (Digital Surf, Besançon, France) was used to enable 3D stereoscopic reconstruction and characterize surface roughness by calculating the arithmetic average of 3D roughness (Sa) according to calculation conditions defined in ISO25178. In case of a highly uneven surface, Mountains SEM filters were used to level and normalize surface data prior to analysis.

**Statistical analysis**. Statistical analysis was conducted using the GraphPad Prism software package, version 8.0 (San Diego, USA). Descriptive analyses of values (D'Agostino-Pearson or Shapiro–Wilk normality tests) were performed before each evaluation to select the appropriate statistical test. Kruskal–Wallis test followed by Dunn's multiple comparison correction were used to compare ovarian tissue differences between age groups, including fiber thickness, pore number, elasticity, and viscoelasticity. One-way analysis of variance followed by Tukey's post hoc multiple comparisons were applied to compare fiber straightness, pore area, surface roughness (Sa), and angle frequency of fiber orientation around different follicle stages within the same age group. Differences in fiber remodeling around follicles at the same follicle stage between prepubertal and reproductive-age tissue were determined by comparing their angle frequencies using the paired t test. Circular statistics[71] were used to compare global collagen fiber orientation at prepuberty, reproductive age, and menopause. The Rayleigh test was applied using the Matlab CircHist toolbox[63] to assess whether fibers were uniformly oriented from 0° to 180° within each age group or had a common mean direction[72]. Watson's parametric multiple sample test and Kuiper's test were carried out using the circular statistics toolbox from Matlab[73] to successively compare mean directions and directional distributions between study groups. Differences were considered significant at values of $p < 0.05$.

**Reporting summary**. Further information on research design is available in the Nature Research Reporting Summary linked to this article.

## Data availability

All analytical data associated with this study are available in the main text, Supplementary Information, and Source Data file. Raw data of AFM analysis can be accessed via: https://data.mendeley.com/datasets/4mcb999nch/draft?a=0c2ad3ac-9c21-4f9c-9779-9202b17e9993 [74]. Source data are provided with this paper.

## Code availability

The Matlab code used in this study, unless otherwise stated in the text, can be found at: https://github.com/inatamara/AFManalysisMatlab [70].

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

## Acknowledgements

The authors thank Corry Charlier (Electron Microscopy Department, University of Namur) for his technical assistance during SEM image acquisition and François Petiteau (Digital Surf) for his help with the Mountains SEM software. We also thank Vincent Fleury for moderating the established collaboration to conduct AFM analyses and Didier Vertommen for his critical review of the manuscript from a biochemical perspective. We greatly thank Yuming Liu for sharing his expertise on using CT-FIRE. We are also very grateful to Mira Hryniuk, BA, for reviewing the English language of the manuscript. This study was supported by grants from the Fonds National de la Recherche Scientifique de Belgique (FNRS) (C.A.A. is an FRS-FNRS Research Associate; grant MIS #F4535 16 awarded to C.A.A.) and the Université Catholique de Louvain (PhD grant "Coopération au développement" awarded to E.O.).

## Author contributions

Conceptualization: E.O. and C.A.A. Methodology, validation, data analysis, and investigation: E.O., O.V.K., A.P., and K.T.H. Software: E.O., A.P., and K.T.H. Resources: T.T., M.O., C.A.A., and M.-M.D. Writing—original draft: E.O. Writing—review and editing: E.O., A.P., K.T.H., O.V.K., M.-M.D., and C.A.A. Data visualization: E.O. Supervision: C.A.A. Funding acquisition: C.A.A. and E.O.

## Competing interests

The authors declare no competing interests.
