## [Peer Review File · Nature Communications]

A blueprint of the topology and mechanics of the human ovary for next-generation bioengineering and diagnosisReviewers' Comments:

Reviewer #1:

Remarks to the Author:

The manuscript by Ouni et al. sought to investigate the mechanical properties, extracellular matrix (ECM) porosity, global fibril orientation, and nanoscale and mesoscale changes human ovarian tissue during reproductive lifespan of the ovarian cortex. This study is the first to rigorously address the lack of information regarding human ovarian ECM and mechanical properties. Experiments included: fiber and pore characterization using scanning electron microscopy at 3 different magnifications (5,000X, 12,000X and 20,000X); Sirius Red stains for collagen fiber orientation analysis; atomic force microscopy to assess elasticity and viscoelasticity; and topography measurement. Ovarian cortex was studied from prepubertal, reproductive age, and menopausal women. The results demonstrated a unique ECM architecture at each reproductive age. For instance, Young's Modulus was approximately 3178 Pa (\pm 245) for reproductive age ovaries but was significantly more rigid for prepubertal and menopausal ovaries. Also, in reproductive age ovaries the fibers of intermediate diameter were assembled into thickest bundles compared to prepubertal and menopausal tissues. Additionally, analysis of collagen fiber orientation around the borders of preantral ovarian follicles at both prepuberty and reproductive age showed a dramatic reorganization of their fibrous microenvironment at each follicle stage. Lastly, the authors found the softer ovarian tissue at reproductive age was characterized by a smoother surface than rougher prepubertal and perimenopausal ovaries. The authors conclude, "...our study provides the first conclusive proof of a link between ECM rigidity and fertility by comparing different stages of ovarian transformation related to a woman's reproductive life." In general, the manuscript is well, written, the figures informative and the results believable. The statistical analysis is rigorous.

General comment: The "question of questions" before the field of reproductive medicine is: how are human primordial follicles maintained in their quiescent state? This question is of fundamental importance for understanding the processes that regulate fertility in our species. Answers to this question could inform treatments for millions with infertility, preservation of fertility for millions of cancer patients, treatment of common conditions (such as PCOS), contraception, and even aging. Unfortunately, maintenance of primordial follicle arrest was incompletely understood. Notably, the ovarian ECM is very dynamic and the process of gametogenesis in the female is accompanied by monthly increases in follicle size from microns to centimeters (a 700-fold size increase). Moreover, results in the past several years have suggested that mechanical signaling may play a role in primordial follicle arrest, but until this report, there were no rigorous mechanical analyses of the ovarian tissue changes from prepubertal through reproductive to menopausal age. This report is indeed "...the first conclusive proof of a link between ECM rigidity and fertility by comparing different stages of ovarian transformation related to a woman's reproductive life." This report provides key evidence that ECM organization and mechanical signaling in the ovary does play a role in maintenance of primordial follicle arrest. Strengths are the rigor of the approach, novelty of the application to this condition and importance of mechanical signaling to oogenesis. Weaknesses are negligible. This report represents an important contribution to the field of reproduction and is a significant advance in understanding. This report has wide ranging implications for other fields, such as tissue engineering, oncology and aging. The data will inform biomimetic scaffolds for a tissue-engineered ovary, aid understanding of ovarian pathology, and direct research for ovarian tissue preservation for cancer and fertility preservation. The work is sure to be widely cited.

Specific suggestion:

The supplemental Figure 2 and SFig 3B might be included in the primary figure set for the reader's benefit.

Reviewer #2:

Remarks to the Author:

This manuscript submitted by Ouni et al. described the differences in the architecture, topography and possible mechanics between pre-pubertal, reproductive-age, and menopausal ovarian tissues. Based on the findings, they proposed a unique biophysical phenotype of reproductive-age tissues,

hoping to bridge biophysics and ovarian function. This manuscript is well written but the significance may be oversold according to the data provided. Nevertheless, the findings are of interest and not many research groups are able to perform this type of detailed study for the first time in human ovaries. Several concerns are summarized below.

1. The ECM architecture in ovarian tissues in reproductive women may vary according to luteal vs. proliferative cycle. It was said in the paper, "Therefore, only fertile patients under ovarian contraceptive treatment...was included." This study missed this critical point.
2. It is not surprising to detect the difference in fibrous network morphology as shown, but what is unclear is the implications and significance of these findings from the perspective of functional biology. The paper did not provide any evidence or articulate the previous studies in details (from other tissue types) how the ECM architecture changes affect tissue functions. It is a pure descriptive study without much biological impact.
3. The authors may deliberate more on the significance of fiber orientation in the peri-follicular ECM around pre-antral follicles.
4. The ovarian ECM can be heterogeneous and sampling bias for the biophysical studies can be a concern.
5. Is there any correlation between the biophysical findings and morphology on light microscopic levels?

Reviewer #3:

Remarks to the Author:

The authors report on architecture and mechanical properties of human ovary tissue aiming at determining differences between healthy prepubertal, reproductive-age, and menopausal ovarian tissues. Although the topic per se is interesting, the role and importance of the study is weakly convincing. These studies are not hypothesis-driven - they rather report on obtained results. In addition, there are some methodological doubts that question the appropriateness and usefulness of the applied methodology. Statistics lack of basic information - how many points, fibres, etc. were analyzed to calculate a particular measure. Statistical significance is dependent on the number of studied samples. Taking into account that the authors analyzed samples from only 5 patients the large statistical significance may not reflect the real difference occurring between patients. Therefore, giving exact numbers is very important. The presented results are not sufficient to draw a conclusion. The novelty of the study is weakly highlighted. Therefore, I am not recommending this manuscript to be published in Nat. Comm. In my opinion, the manuscript is not suitable for this journal:

Specific remarks:

- 1) Fig.1 - the authors present the results of diameter assessment for fibrils, fibres and fibre bundles. What denotes n? Number of samples or the number of fibrils, fibres and fibre bundles. This has to be specified, in particular for fibres and fibrils.
- 2) Although, the authors defined in Fig.1a what means fibrils, fibres and fibre bundles, it is not clear how they recognized these structures from SEM images.
- 3) The authors are comparing samples at fibre bundle (5000 x) and fibre (20000 x) scales. Why such parameters such as fibre diameter, number of pores and pore area is dependent on the image magnification? Even if, the relations between these parameters determined at three stages: prepuberty, reproductive age and menopause tissues should be the same e.g. if for 5000x fibre diameter follows: $D_{\text{prepuberty}} < D_{\text{menopause}} < D_{\text{reproductive age}}$ the similar relation should be visible when higher magnification images are analyzed. Otherwise, the study requires for larger statistics. For pore diameter is OK as higher magnification can reveal smaller pores, while the pore diameter should be independent of the image resolution. Such discrepancy presented by authors may denote the large variability among the same tissue sample and require to collect more images from the same sample.
- 4) It seems that attributing fibres alignment bases on histological assessment of elastin and

collagen. It should be clearly stated (in the section "Interstitial fibre orientation and straightness change with age and hormonal state") why suddenly the authors are saying "local collagen alignment" after first section in which the origin of fibres is unknown.

5) In supplementary Fig. S1, the authors introduced a "fibre length". How the length is defined. In the included SEM images the fibres are "cut" by the choosing the image size. This question the use of length as a parameter that can quantify the tissue structure.

6) Fiber orientation - again that authors are not specifying how many fibres were analyzed (what denotes two blue starts in Fig.3B, inside radial graphs?). The statistical difference can be significant if there is a larger number of fibres analyzed. If a few fibres were analyzed then there will not be statistical difference.

7) AFM studies - here I have major doubts.

a) water is not a physiological solution thus due to different osmolarity may destroy the tissue. In AFM studies, washing in water is not advisable as it may destroy the tissue and alter its mechanics.

The results showing tissue treated with and without water should be presented.

b) the authors present the nominal spring value - what was a variability of the cantilever spring constant. How many cantilevers were used in the measurements?

c) did Dimitriadis correction was applied?

d) what was the indentation dept analyzed.

e) force curves should be shown together with moduli distributions, especially that distributions are not symmetric and using means +/- standard deviations is not the best way.

Median is better. Mean was calculated from all force curves or force maps?

f) how many maps were recorded? It seems that 3 per each tissue - this is not sufficient. What is the heterogeneity of an individual sample?

g) why relaxation times are shown without error? They display mean or median?

h) How many curves/maps were recorded to assess the viscoelasticity?

f) what is the physical meaning of fast "tau" and slow "tau". What model was used to determine these values?

8) Roughness should be correlated with SEM images. How many images were used to calculate Sa value? What is the importance of using 10 um x 10 um topography images (and roughness) as compared to SEM images that better demonstrate the age-dependent difference in ovary tissue?

9) The authors do not cite any tissue-oriented AFM studies. This is important to evaluate the quality of the obtained results in terms of tissue mechanics. This unfortunately explain weak points in AFM-based analysis of ovarian tissue. Some references:

Puttini et al. Mol. Therapy 2009 - muscle tissue

Plodinec et al. Nature Nanotechnol 2012 - breast cancer

Lekka et al. Arch. Biochem.Biophys 2012 - breast, vulvar, endometrium

Tan et al. 2015 Nanoscale - liver tissue

Bouchonville et al. Soft matter 2016 - brain tissue

Ciasca et al. Nanoscale 2016 - brain tissue

Anura et al. J. mech. Behav. Biomed. Devices 2017 - epithelial connective tissue

Deptula et al. ACS Biomater. Sci. Eng. 2020 - colon cancer

Calibration is well described in Schillers et al. Sci. Reports 2017.

Reviewer #1:

The manuscript by Ouni et al. sought to investigate the mechanical properties, extracellular matrix (ECM) porosity, global fibril orientation, and nanoscale and mesoscale changes human ovarian tissue during reproductive lifespan of the ovarian cortex. **This study is the first to rigorously address the lack of information regarding human ovarian ECM and mechanical properties.** Experiments included: fiber and pore characterization using scanning electron microscopy at 3 different magnifications (5,000X, 12,000X and 20,000X); Sirius Red stains for collagen fiber orientation analysis; atomic force microscopy to assess elasticity and viscoelasticity; and topography measurement. Ovarian cortex was studied from prepubertal, reproductive age, and menopausal women. The results demonstrated a unique ECM architecture at each reproductive age. For instance, Young's Modulus was approximately 3178 Pa (\pm 245) for reproductive age ovaries but was significantly more rigid for prepubertal and menopausal ovaries. Also, in reproductive age ovaries the fibers of intermediate diameter were assembled into thickest bundles compared to prepubertal and menopausal tissues. Additionally, analysis of collagen fiber orientation around the borders of preantral ovarian follicles at both prepuberty and reproductive age showed a dramatic reorganization of their fibrous microenvironment at each follicle stage. Lastly, the authors found the softer ovarian tissue at reproductive age was characterized by a smoother surface than rougher prepubertal and perimenopausal ovaries. The authors conclude, "...our study provides the first conclusive proof of a link between ECM rigidity and fertility by comparing different stages of ovarian transformation related to a woman's reproductive life." In general, the manuscript is well written, the figures informative and the results believable. The statistical analysis is rigorous.

General comment: The "question of questions" before the field of reproductive medicine is: how are human primordial follicles maintained in their quiescent state? This question is of fundamental importance for understanding the processes that regulate fertility in our species. Answers to this question could inform treatments for millions with infertility, preservation of fertility for millions of cancer patients, treatment of common conditions (such as PCOS), contraception, and even aging. Unfortunately, maintenance of primordial follicle arrest was incompletely understood. Notably, the ovarian ECM is very dynamic and the process of gametogenesis in the female is accompanied by monthly increases in follicle size from microns to centimeters (a 700-fold size increase). Moreover, results in the past several years have suggested that mechanical signaling may play a role in primordial follicle arrest, but until this report, there were no rigorous mechanical analyses of the ovarian tissue changes from prepubertal through reproductive to menopausal age. This report is indeed "...the first conclusive proof of a link between ECM rigidity and fertility by comparing different stages of ovarian transformation related to a woman's reproductive life." This report provides key evidence that ECM organization and mechanical signaling in the ovary does play a role in maintenance of primordial follicle arrest. Strengths are the rigor of the approach, novelty of the application to this condition and importance of mechanical signaling to oogenesis. Weaknesses are negligible. This report represents an important contribution to the field of reproduction and is a significant advance in understanding. This report has wide ranging implications for other fields, such as tissue engineering, oncology and aging. The data will inform biomimetic scaffolds for a tissue-engineered ovary, aid

understanding of ovarian pathology, and direct research for ovarian tissue preservation for cancer and fertility preservation. The work is sure to be widely cited.

We thank Reviewer#1 for his/her highly positive comments and appreciation of the importance and perspectives of our work in the field.

Specific suggestion:

1-The supplemental Figure 2 and SFig 3B might be included in the primary figure set for the reader's benefit.

Supplemental Figures 2 and 3b have been included in the primary figure set, as requested.

Reviewer #2:

This manuscript submitted by Ouni et al. described the differences in the architecture, topography and possible mechanics between pre-pubertal, reproductive-age, and menopausal ovarian tissues. Based on the findings, they proposed a unique biophysical phenotype of reproductive-age tissues, hoping to bridge biophysics and ovarian function. This manuscript is well written but the significance may be oversold according to the data provided. Nevertheless, the findings are of interest and not many research groups are able to perform this type of detailed study for the first time in human ovaries.

We thank Reviewer#2 for his/her comments. For the sake of being concise, we did not include an extensive discussion on the importance of our findings. However, we believe that our results represent a landmark in the analysis and understanding of the human ovarian ECM. Reviewer #1, who is clearly an expert in the field of reproductive medicine, could see that our study "provides key evidence that ECM organization and mechanical signaling in the ovary does play a role in maintenance of primordial follicle arrest" and "represents an important contribution to the field of reproduction and is a significant advance in understanding". Moreover, he/she highlighted the potential applications of our findings, stating that our data have "wide ranging implications for other fields, such as tissue engineering, oncology and aging, and will inform biomimetic scaffolds for a tissue-engineered ovary, aid understanding of ovarian pathology, and direct research for ovarian tissue preservation for cancer and fertility preservation." Nevertheless, since the significance of our findings was not clear to Reviewer #2, we have modified the manuscript discussion and addressed this point and all other reviewer concerns.

Several concerns are summarized below.

1. The ECM architecture in ovarian tissues in reproductive women may vary according to luteal vs. proliferative cycle. It was said in the paper, "Therefore, only fertile patients under ovarian contraceptive treatment...was included." This study missed this critical point.

We agree that it would be interesting to investigate the hypothesis that the cycle phase may have an impact on the ECM architecture. However, collecting the required number of ovarian samples from healthy women (not using hormone-based contraceptives) undergoing a laparoscopic procedure during the luteal or follicular phase would demand years. Moreover, since it is rarely necessary to assess and record the cycle stage of patients, this information is hardly ever found in their medical files. On the other hand, a huge number of fertile patients are under contraceptive treatment and they must disclose this information during the consultation. In order to avoid any heterogeneity within the same group, we decided to focus only on fertile patients taking hormone contraceptive treatment.

Considering that in today's society the majority of women use hormone-based contraceptives, as described by the World Health Organization (United Nations, Department of Economic and Social Affairs, Population Division. Trends in contraceptive use worldwide 2015 (ST/ESA/SER.A/349)), our study describes the ovarian ECM architecture of the most significant subgroup of the modern female population. Moreover, to the best of our knowledge, a direct effect of estroprogestatives on the ovarian ECM has never been reported, especially from a biomechanical perspective, so this represents additional originality of our study.

It is also important to stress that in their reference map of the human ovary, Wang et al. (*J Mol Med*, 2005) reported very limited proteomic differences between ovarian tissue at luteal and proliferative stages from a biochemical point of view, and these limited differences did not belong to the ECM. This may indicate that our sampling is representative of the ovarian state of women at reproductive age. In addition, the fact that missing just one pill puts patients at high risk of pregnancy demonstrates that oral contraceptives probably do not target the ovarian ECM, but rather endocrine functions.

We have modified the manuscript discussion to suggest investigating the hypothesis raised by the reviewer.

L391-405:

In order to avoid any heterogeneity within the same group, we decided to focus only on fertile patients taking hormone contraceptive treatment. Since most adult women in the consultation practice declared to use such a type of contraceptive strategy, it was faster to collect tissue samples from them. Indeed, considering that in today's society the majority of women use hormone-based contraceptives, as described by the World Health Organization (ST/ESA/SER.A/349), our study shows the ovarian ECM architecture of the most significant subgroup of the modern female population. Moreover, to the best of our knowledge, a direct effect of estroprogestatives on the ovarian ECM has never been reported, especially from a biomechanical perspective. On the other hand, it would be interesting to investigate if the cycle phase could influence the ECM architecture. However, such a study would be much more challenging as collecting the required number of ovarian samples from healthy women (not using hormone-based contraceptives) undergoing a laparoscopic procedure during the luteal or follicular phase would demand years. Moreover, since it is rarely necessary to assess and record the cycle stage of patients, this information is hardly ever found in their medical files.

2. It is not surprising to detect the difference in fibrous network morphology as shown, but what is unclear is the implications and significance of these findings from the perspective of functional biology. The paper did not provide any evidence or articulate the previous studies in details (from other tissue types) how the ECM architecture changes affect tissue functions. It is a pure descriptive study without much biological impact.

When preparing the manuscript, we decided not to extend our discussion by comparing our findings with data from the literature. However, as requested by the reviewer, we have amended and completed the manuscript with the paragraphs below.

L190-244:

Along with the cell niche, the ECM has been shown to orchestrate cell behavior and fate ¹⁸. For instance, the multiscale architecture of ECM collagen fibers influences cell polarity and promotes migration by providing contact guidance cues ¹⁹. While a normal ECM can restore transformed cells to quiescence ¹⁸, a damaged ECM can trigger malignancy ²⁰ or differentiation ²¹. ECM architecture plays a role in the specialized function of the tissue of which it is a part ²². Indeed, collagen forms thick fibers in load-bearing tendons that are aligned along the tendon to optimize force transmission and tendon strength ²³. By contrast, collagen in the cornea forms woven sheets of thin fibers that provide strength combined with optical

transparency²³. In interstitial tissue, collagen forms mostly isotropic networks, which provide mechanical strength combined with porosity to facilitate nutrient transport and cell migration.

While secretory patterns of hormones and paracrine and autocrine functions in the human ovary have been investigated for several decades, little is known about the role of ECM biomechanics and topology in ovarian function. Since its first description more than a century ago, the human ovary has been known to have a dynamic architecture, with a stiffer collagen-rich cortex and a softer medullary layer. Follicles are distributed along this collagen gradient: early preantral follicles are located in the firm cortex and late secondary and antral follicles are found in the less rigid medullary layer. Only recently, studies highlighted the fact that the mechanical properties of the ovarian ECM, including this rigidity gradient, play a crucial role in supporting follicle survival and folliculogenesis²⁴. Studies are finally revealing that the ovary is a dynamic mechano-responsive organ and that crosstalk between the ECM and follicles and the ECM and ovarian cells is essential for folliculogenesis and oogenesis. Indeed, ovulation would never occur without remodeling of the dense and rigid ECM of the ovarian cortex. The 'weakening' of the ECM by enzymes synthesized by cells from preovulatory follicles is vital for oocyte expulsion²⁵. Another example of the influence of ECM composition and architecture on follicle development has been seen in subjects with polycystic ovary syndrome (PCOS). Ovaries from PCOS patients have a densely collagenized and thickened cortex that probably creates a biomechanically non-permissive environment, possibly altering mechanical signaling. This environment is likely to play a role in boosting numbers of growing follicles through a cascade of events, culminating in increased secretion of growth factors, which leads to primordial follicle activation and development²⁶. PCOS ovaries are also characterized by anovulation, probably a result of the lack of degradation of the fibrous ECM¹¹ or an undiagnosed abnormal ECM architecture.

Kawamura et al.²⁷ reported an increase in primordial follicle activation in PCOS patients after cutting ovarian tissue into small cubes. This procedure was found to modify ovarian mechanical forces by releasing tension on ovarian cells and disrupting the Hippo signaling pathway, characterized by polymerization of globular actin to filamentous actin. Actin polymerization may connect biophysical changes with suppression of Hippo signaling, increasing nuclear YAP concentrations and stimulating follicle growth²⁸. Moreover, Telfer and McLaughlin (patent WO2014043835A1) proved that stretching small strips of cortical ovarian tissue in one direction by more than 10% of the initial length prior to in vitro culture enhances primordial follicle activation to the secondary stage. These initial observations highlight the role of ECM architecture as a regulator of cellular/follicle behavior in the ovary, demonstrating the importance of microenvironmental cues to female fertility. Although mechanical stimulation is mediated by specific architecture in different tissues²⁹, there is still a lack of understanding of ovarian ECM topology that needs to be addressed. We therefore believe that identifying differences in the ECM architecture of ovaries before and after puberty and during menopause will help us elucidate how biochemical regulation can be correlated to ECM morphology. Our spatiotemporal follow-up of fibrous ECM morphology provides novel phenotyping of ovarian tissue at different ages, shedding light on the possible effect of the fibrillar structure on the signaling

of growth factors binding to the fiber surface. Here, the ECM brings active molecules into close proximity with the cell surface and facilitates interactions between growth factors and integrins, or sequesters them within the fibrous network and tight pores ³⁰.

3. The authors may deliberate more on the significance of fiber orientation in the perifollicular ECM around pre-antral follicles

In response to this suggestion, we have amended the manuscript with the paragraph below.

L276-318:

Our results complement and corroborate in vitro studies demonstrating that regulation of ovarian follicle development depends on the architecture of the perifollicular ECM, in addition to endocrine- and paracrine-acting hormones ³⁷. *Beyond providing structural support for follicle formation and growth, the fibrous ECM network acts as a reservoir for paracrine and endocrine signals inside the ovary and permits or restricts their access to cells within follicles. Similarly to other organs, the ovarian ECM is prone to mechanical and enzymatic remodeling in different physiological and pathological conditions* ³⁷. *Hence, our observation of differences between perifollicular ECM architecture in prepubertal and reproductive-age follicles at similar stages could be a cause of differences in molecular signaling and interfollicular communication, responsible for the life cycle of preantral follicles. Throughout folliculogenesis, ECM architecture ensures appropriate hormone secretion, somatic cell differentiation, and oocyte maturation* ¹¹. *In conjunction with this activity, the ovary accommodates a large reserve of inactive primordial follicles that contain nongrowing oocytes and nondividing, flattened (squamous) pregranulosa cells surrounded by a basal lamina. Following entry into the growing follicle pool upon activation, squamous pregranulosa cells surrounding the oocyte become cuboidal granulosa cells (primary follicles). They proliferate to form multiple layers, a morphological hallmark of secondary follicles. At this stage, the theca cell layer is recruited from the stroma to surround the basal lamina, which involves intricate polarity changes and cell migration processes. Fortunately, advances in in vitro follicle culture have allowed us to decipher the role of ECM architecture in follicle and ovarian development. It was found that cell adhesion, supported by the ECM, induces changes in cell shape and motility necessary for various cellular functions during folliculogenesis. A number of studies have also shown that in vitro culture of granulosa cells plated on matrices of ECM proteins of various densities also leads to alterations in cell adhesion and shape* ³⁸, *highlighting the importance of using biomimetic scaffolds to achieve completion of folliculogenesis in vitro. This can only be accomplished by following our blueprint of fiber orientation and angular density of the perifollicular ECM, and our previous description of elastic matrix spatiotemporal changes* ¹⁷. *Our results also revealed that at all stages of folliculogenesis, prepubertal follicles are surrounded by a higher density of fibers. This can predict a non-permissive environment at a very early stage of preantral development, as already suggested* ³⁹, *or peculiar architectural regulation of molecular signaling and transmission activity, as described above, which both play key roles in follicle quiescence and ovarian reserve preservation during*

prepuberty. Moreover, follicle survival has been described in vitro as dependent on scaffold geometry, where greater adhesion of follicles to ECM fibers oriented below 90° resulted in the highest survival rates ⁴⁰. By comparing local fiber directionality around follicle borders at primordial, primary and secondary stages, we noted that before and after puberty, secondary follicles appear to modify their microenvironment arrangement locally compared to follicles at earlier stages of development by reorienting the majority of collagen fibers below 50°. This could indicate that follicles at this stage require a higher degree of fiber contact and adhesion signaling ³⁶ to thrive and complete their development and maturation towards ovulation ⁴¹.

4. The ovarian ECM can be heterogeneous and sampling bias for the biophysical studies can be a concern.

We agree with the reviewer and, for this reason, in every ovarian sample (cortex) in each group, at least three 100 x 100 μm² force maps and nine positions each followed by viscoelastic measurement (force spectroscopy) at two points in the same tested region. Using this sampling method, we did not note any significant heterogeneity. Moreover, the consistency and reproducibility of our measurements in each group, as demonstrated by box plots and statistical analyses, further highlight the absence of sampling bias in our procedure. See below an example of three regions mapped within the same sample:

5. Is there any correlation between the biophysical findings and morphology on light microscopic levels?

In our study, we correlated our biophysical findings with elastic matrisome component analysis under fluorescent microscopy, as demonstrated in Supplementary Figures S2 and S3. Since architecture is a composite of many different features and scales, it was difficult to correlate it to our biophysical findings. We have also discussed the contribution of the differences in architecture and porosity between groups in their specific mechanical phenotypes. Nevertheless, no correlation could be established between the biophysical findings and morphology by light microscopy.

Reviewer #3:

The authors report on architecture and mechanical properties of human ovary tissue aiming at determining differences between healthy prepubertal, reproductive-age, and menopausal ovarian tissues. Although the topic per se is interesting, the role and importance of the study is weakly convincing.

We regret that the role and importance of our study was not clear to Reviewer #3. Indeed, in the field of reproductive medicine, our results represent a landmark in the analysis and understanding of the human ovarian ECM. Reviewer #1, who is clearly an expert in reproductive medicine, could see that our study "provides key evidence that ECM organization and mechanical signaling in the ovary does play a role in maintenance of primordial follicle arrest" and "represents an important contribution to the field of reproduction and is a significant advance in understanding." Moreover, he/she highlighted the potential applications of our findings by stating that our data have "wide ranging implications for other fields, such as tissue engineering, oncology and aging, and will inform biomimetic scaffolds for a tissue-engineered ovary, aid understanding of ovarian pathology, and direct research for ovarian tissue preservation for cancer and fertility preservation."

Similarly to any other tissue, the ECM provides structure and regulates numerous cellular functions in the ovary. For several decades, we have been investigating the biochemical signals involved in folliculogenesis and oogenesis, but only recently have we uncovered the role of the physical properties of the ovarian ECM, such as matrix elasticity and architecture, in these processes. In recent years, studies have demonstrated that these ECM properties provide essential instructive cues regulating follicle fate. For instance, Woodruff's groups showed that in vitro culture of murine follicles in stiff matrices would maintain follicle dormancy, whereas less rigid matrices would produce larger, more hormonally productive follicles, with higher fertilization rates (West et al. Biomaterials, 2007; Xu et al. Biol Reprod, 2006). Du et al. (Reproduction, 2008) reported that high hydrostatic pressure treatment of porcine oocytes, in vitro-matured prior to vitrification, would significantly increase their blastocyst formation rate. More recently, approaches have been proposed to exploit the responsiveness of ovarian tissue to mechanical signals for reproductive purposes. Kawamura et al. (PNAS, 2013) reported that reinforcing disruption of the Hippo signaling pathway in fragmented cortical ovarian tissue using treatment with PI3K stimulators and PTEN inhibitors (both of which activate primordial follicles) yielded healthy offspring in women with primary ovarian insufficiency (POI). Additionally, Telfer and McLaughlin (patent WO2014043835A1) proved that stretching small strips of cortical ovarian tissue in one direction by more than 10% of their initial length prior to in vitro culture enhanced primordial follicle activation to the secondary stage.

These initial observations highlight the role of the ECM as a regulator of cellular/follicle behavior in the ovary, demonstrating the importance of microenvironmental cues to female fertility. However, in all approaches, a lack of knowledge of the biomechanical behavior of cortical ovarian tissue in vivo and poor quantitative characterization of the mechanical challenges do not allow us to correctly estimate the extent of mechanical stimulation and the possible remodeling of ECM with age, which ultimately hampers reproducible and elective treatments.

Most importantly, our study provides key elements to explain how human primordial follicles are maintained in their quiescent state by focusing on ECM topology and

mechanics, a field which is greatly underinvestigated, particularly in human reproduction. This question is of fundamental importance to elucidating processes that regulate fertility in humans. Answers to this question could help us understand the conditions leading to subfertility or infertility, such as PCOS, and develop tailored treatments. Moreover, it may guide us towards development of novel strategies for contraception and menopause. Until now, the real involvement of biomechanics and architecture in ovarian reserve maintenance and ovarian maturation and aging was not fully understood. Our study is the first conclusive proof of a link between ECM rigidity and fertility, acquired by comparing different stages of ovarian transformation related to a woman's reproductive life. This is especially important when considering that aging is the prominent hallmark of the ovarian reserve and fertility, and the reproductive-age ovary serves as a gold standard for a functional biomimetic ovary as a fertility restoration solution.

This report represents an important contribution not only to the field of reproduction, but also to tissue engineering and regenerative medicine, as also highlighted by Reviewer #1. Finally, as pointed out by Reviewer #2, "not many research groups are able to perform this type of detailed study for the first time in human ovaries".

These studies are not hypothesis-driven - they rather report on obtained results.

Our hypothesis is that the human ovary has unique physical and topological ECM features at reproductive age, tailored to its biological function and playing an active role in its fertility and ovarian activity compared to prepubertal and menopausal tissue, which are known for their quiescent follicles and ovarian inactivity. We have modified the manuscript to clearly state our hypothesis.

L68-72:

Here, we directly investigate the changes to human ovarian tissue during its lifespan to confirm our hypothesis that the human ovary has unique physical and topological ECM features at reproductive age. Such properties are tailored to ECM biological function and play an active role in fertility and ovarian activity compared to prepubertal and menopausal tissue, which are known for their quiescent follicles and ovarian inactivity.

In addition, there are some methodological doubts that question the appropriateness and usefulness of the applied methodology. Statistics lack of basic information - how many points, fibres, etc. were analyzed to calculate a particular measure. Statistical significance is dependent on the number of studied samples. Taking into account that the authors analyzed samples from only 5 patients the large statistical significance may not reflect the real difference occurring between patients. Therefore, giving exact numbers is very important. The presented results are not sufficient to draw a conclusion. The novelty of the study is weakly highlighted. Therefore, I am not recommending this manuscript to be published in Nat. Comm. In my opinion, the manuscript is not suitable for this journal:

All the issues raised by the reviewer have been answered point-by-point, as described below.

Specific remarks:

1) Fig.1 - the authors present the results of diameter assessment for fibrils, fibres and fibre bundles. What denotes n? Numer of samples or the number of fibrils, fibres and fibre bundles. This has to be specified, in particular for fibres and fibrils.

The "n" denotes the number of biological samples (patients) analyzed. This was clarified in the new version of the manuscript, as shown below:

L814-821:

Figure 1. ECM microstructure and fibrous network morphology in human ovarian tissue from prepuberty to menopause. A) Schematic illustration of the fibrous network anatomy composed of fibrils, fibers and fiber bundles, as defined in this study. SEM micrographs revealed: B) the ECM network structure at fiber bundle scale (5,000X magnification) and C) fiber scale (20,000X magnification). D, E) At prepuberty (number of biological replicates [n]=5), ovarian tissue is composed of the thinnest fibers (mean \pm SD: 76.1 nm \pm 9.8) assembled into the thinnest bundles (160.0 nm \pm 21.3), densifying upon puberty (fiber scale: 528.4 nm \pm 128.0; fiber bundle scale: 3,379.0 nm \pm 368.8).

2) Although, the authors defined in Fig.1a what means fibrils, fibres and fibre bundles, it is not clear how they recognized this structures from SEM images.

Fiber bundles could be observed at 5,000X magnification thanks to their characteristic fiber assembly. When we zoom into these bundles, we can reach their elementary fibers and observe their spacing (pores) (Fig. 1, 2). Such a unique and straightforward approach enables us to capture multiscale architectural features. This is explained in the manuscript, as shown below:

L88-94:

In this study, we define fibrils as individual entities with discrete diameters in the order of several nanometers, whereas fibers are defined as a pack of multiple fibrils that can be assembled into bundles (Fig. 1A). While it is rare to observe isolated fibrils in ovarian cortex (Fig. 1B), we were able to capture the main features of its fibers at 20,000X magnification and their arrangement into bundles at 5,000X magnification (Fig. 1, B and C). After identifying fiber bundles at 5,000X, we zoomed in to pinpoint the features of their elementary fibers at 20,000X (Fig. 2).

3) The authors are comparing samples at fibre bundle (5000 x) and fibre (20000 x) scales. Why such parameters such a fibre diameter, number of pores and pore area is dependent on the image magnification? Even if, the relations between these parameters determined at three stages prepuberty, reproductive age and menopause tissues should be the same e.g. of for 5000x fibre diameter follows: $D_{\text{prepuberty}} < D_{\text{menopause}} < D_{\text{reproductive age}}$ the similar relation should be visible when higher magnification images are analyzed. Otherwise, the study requires for larger statistics. For pore diameter is OK as higher magnification can reveal smaller pores, while the pore diameter should be independent of the image resolution. Such discrepancy presented by authors may denote the large variability among the same tissue sample and require to collect more images from the same sample.

Fiber bundles and fiber diameters are different, since fiber bundle diameter is defined by the diameter of its elementary fibers and also by their spacing and tightness. Accordingly, while pores at 5,000X represent spacing between bundles, at 20,000X they represent spacing between fibers. At different magnifications, we can see different elements of the architecture of the ovarian ECM.

As described in the methods section, at each magnification and from each sample (n=5/group), we acquired at least 3 regions by SEM, which led to highly significant results with reasonable variability.

We understand the reviewer's confusion, since in Figure 1D we noticed a mistake in the axis name, where fiber diameter should have been fiber bundle diameter. We apologize for this mistake and have corrected it.

P31:

4) It seems that attributing fibres alignment bases on histological assessment of elastin and collagen. It should be clearly stated (in the section "Interstitial fibre orientation and straightness change with age and hormonal state") why suddenly the authors are saying "local collagen alignment" after first section in which the origin of fibres is unknown.

We understand the confusion as the word 'local' in this section is ambiguous. We have changed the description of collagen alignment.

L116-119:

All age groups showed directional collagen alignment, as demonstrated by fiber orientation analysis (Supplementary Fig. S1). They all displayed preferred average fiber orientation centered around 90°: 89.53° ± 0.33 at prepuberty; 91.63° ± 0.9 at reproductive age; and 89.14° ± 0.27 at menopause (Fig. 3B).

5) In supplementary Fig. S1, the authors introduced a "fibre length". How the length is defined. In the included SEM images the fibres are "cut" by the choosing the image size. This question the use of length as a parameter that can quantify the tissue structure.

First, we would like to stress that fiber length was not used to draw any biological conclusions in the manuscript. We describe it in Supplementary Fig. S1, along with fiber straightness, to illustrate that neither factors influence our angle measurement method, as described in Supplementary Fig. S1.C-D. was clarified in the figure legend, as shown below:

L999-1009:

Supplementary Fig S1. Fiber straightness, length and angle measurement. Fiber straightness and length are two factors that might influence fiber tracking and orientation measurement on CT-FIRE, since long curvy fibers can lead to biased mean displacement. Based on fiber length and straightness measurement in all groups, we demonstrate that angle measurement is not influenced by these factors. A) To check whether fiber straightness impacts angle measurement, summarized statistics of straightness data were plotted against corresponding fiber orientation in each sample. No direct interdependence was noted between the two variables, which demonstrates that fiber straightness did not affect measured angles. The data plot fits a smoothing spline with a lambda value of 0.05. The curves also include the bootstrap confidence region for each fit generated by JMP Pro 14.3.0. B) Fiber length variation with age expressed in pixels. Fiber length was measured using CT-FIRE on Sirius Red-stained slides. The Tukey-Kramer HSD test was used to compare mean fiber lengths (orange), ** $p < 0.0001$. C-D). Examples of mean displacement of a fiber trajectory and angle measurement are shown. C) Trajectory consisting of four XY coordinates; displacement from one coordinate to the next is denoted as $dXY1$, $dXY2$ and $dXY3$. D) The displacements (blue: $dXY1$, $dXY2$ and $dXY3$) are averaged to calculate the mean displacement of the trajectory (black arrow).**

To measure fiber length from Sirius Red-stained sections (the same analyzed surface between groups), we used CT-FIRE, which is a Matlab-based open source software for fiber tracking, segmentation and measurement. Fiber length is calculated as the Euclidean distance traveled along the fiber in CT-FIRE. More specifically, during the fiber tracking process, a number of points on the whole central line of a fiber are located and the distances (Euclidean distance) between the adjacent points are calculated. The sum of these distances along the fiber is defined as the fiber length. In other words, a fiber is divided into a number of fiber segments, whose lengths add up to the total fiber length.

6) Fiber orientation - again that authors are not specifying how many fibres were analyzed (what denotes two blue stars in Fig.3B, inside radial graphs?). The statistical difference can be significant if there is a larger number of fibres analyzed. If a few fibres were analyzed then there will not be statistical difference.

The number of analyzed fibers is now detailed in Table 1, which demonstrates the large number of fibers that were segmented and used in statistical analyses to gain a comprehensive understanding of ovarian architecture remodeling with age. Moreover, all statistical studies conducted here were reviewed and approved by the Louvain Institute of Data Analysis and Modeling in Economics and Statistics from the Université Catholique de Louvain before submission.

We thank the reviewer for notifying us about the blue stars that had not been defined in Fig. 3B. The figure legend was corrected, as shown below:

L865-868:

B) Angular fiber distribution changes with age (blue) demonstrating the anisotropic nature of human ovarian tissue, but average angle changes only at reproductive age (red), and no significant differences were noted between prepubertal and menopausal tissues.

L978-979:

Table1. Number of segmented and tracked fibers in perifollicular and interstitial ECM.

	Interstitial ECM	Perifollicular ECM		
		Primordial	Primary	Secondary
Prepubertal	117,005 fibers 74 images	17,025 fibers 48 follicles*	10,224 fibers 28 follicles	3,659 fibers 8 follicles
Reproductive- age	157,445 fibers 104 images	19,246 fibers 56 follicles	10,781 fibers 31 follicles	2,069 fibers 6 follicles
Menopausal	173,325 fibers 102 images	N/A	N/A	N/A

***One follicle per image was analyzed.**

7) AFM studies - here I have major doubts.

Now that we have provided a detailed description of AFM elastic and viscoelastic measurements, we hope that the reviewer's doubts will be allayed. In our field of expertise (tissue biomechanics: co-authors Kalina Haas & Alexis Peaucelle), AFM-based measurements have been previously described in great detail by ourselves and others. We did not therefore provide full details as a rule, but only refer to our previous work. However, we have added the description in the new version of the manuscript.

L513-593:

The 50- μ m-thick cryosections of ovarian tissue were kept at -80°C until the start of the experiment and prepared as described in the section 'sample preparation for AFM'. Rheological properties were measured with the NanoWizard 1 BioScience (JPK Instruments) AFM operating in force spectroscopy mapping mode, as previously described before ^{64, 65, 66}. Force-indentation curves were collected using a rectangular silicon nitride cantilever with a 0.07 N/m nominal spring constant, and 5- μ m-diameter borosilicate glass spherical particle attached to the tip. A spherical probe was selected because it produces considerable force with minimal damage to the surface, so is ideal for compliant materials like ovarian tissue ⁶⁷. The measured spring constant was very close to the nominal value of 0.07 ± 0.005 N/m using the thermal tune method. This step is designed to transform the laser position signal on the AFM receptor into deformation of the cantilever by evaluating its spring constant in order to translate it into a force. The same cantilever was used for all experiments (Novascan, Ames, IA, USA). The indentation force was set to limit maximum indentation to ~ 0.5 -1 μ m, so the bead-tissue contact area during all tests never exceeded 7.068 μ m². The experiments were performed at room temperature using 1X PBS buffer. To prevent mechanical

modifications arising from tissue damage, the tests were limited to 40 minutes per sample. This included at least three 100 x 100 μm^2 force maps and nine positions each with six repetitions of viscoelasticity measurements. No significant difference was observed between measured areas within the same tissue sample, indicating that the ovarian tissue was homogeneously mechanically averaged over the spatial window of 100 x 100 μm^2 . Further mapping would have led to experimentally induced variability. For a 100 x 100 μm^2 area, we performed 50 x 50 measurements, resulting in 2,500 force-indentation experiments. The tissue sections were immobilized on Superfrost Plus glass slides Adhesion slides (Menzel-Glaser, Germany). Tight interaction between the samples and the glass did not induce local ECM modifications (artificial solvents, tapes and glues could dissolve or otherwise disturb the physiological state of the ECM, leading to local softening affecting AFM measurements).

Prior to tissue measurements, AFM sensitivity and alignment were ensured with an undeformable (compared to the biological sample) empty glass slide in PBS. The difference between cantilever deflection on a rigid surface and the compliant tissue sample illustrates the deformation of the tissue under the bead load. The force-indentation (F - δ) curves can be fitted with a single exponential following the Hertzian contact model. The Hertzian contact model generates the relationship between the applied force, F , and the resulting indentation, δ , allowing the extraction of an apparent Young's modulus, E_a (a for apparent); namely the correlation constant between the force and area of indentation. E_a is a standard measure for soft tissue elastic properties. We considered the tissue to be incompressible (assumed Poisson ratio: 0.5). The E_a of the probed samples was calculated by fitting the contact part of the measured approach force curves to a standard Hertzian model for a spherical indenter (tip) of radius, R ⁶⁸:

$$F = (4E\sqrt{R}) * (\delta^3/2) / (3(1 - \nu^2))$$

$$E = 3(1-\nu^2) F / (4\sqrt{R}) * (\delta^3/2)$$

where $\nu=0.5$ is the Poisson ratio and δ is the indentation depth, calculated by subtracting cantilever deflection from tip displacement. The Hertzian model is typically used to determine contact between two linear elastic bodies. As such, several assumptions had to be checked concerning our biological tissues: i) that they displayed linear elasticity at the scale examined; and ii) that the nonhomogeneity of ovarian tissue (as a composite material) was negligible at the scale examined. Our data are well fitted by the Hertzian model, which is the best approximation for Young's modulus of biological tissues ⁶⁹.

Young's modulus is presented using a violin distribution plot overlaid with mean and median values calculated over F - δ curves (i.e., pixelwise). For topographical reconstructions, the height of each point was determined by the point of contact from the F - δ curve, with each contact point issuing from the same curve used to determine E_a . Stiffness data were projected onto topographical maps using MatLab.

To measure the viscoelasticity of ovarian tissue, we conducted repeated and successive long indentation cycles, followed by partial force release. For indentation portion, force was kept constant, while for the release portion, deformation was constant, allowing us to monitor the evolution of both deformation and force. The viscoelastic measurement (force spectroscopy)

immediately followed elastic measurement (force mapping scan) and was performed in two points of the same tested region according to a force ramp design loop of ~240 s duration: cantilever extension with force increase (4- μm indentation, 100 ms); constant height maintenance at the contact point (20 s); cantilever retraction and force decrease (0.8 μm , 20 ms); and finally constant force maintenance (20 s). Data sampling frequency was set to 4,000 Hz.

Viscoelastic materials retain their shape after deformation, but with a time delay, as shown by the relaxation constant. In slow deformations, we can assume that the ECM is incompressible (Young modulus measurements). However, fast viscoelasticity measurements demonstrate that the tissue may be (reversibly) compressible in short time scales. In this context, viscoelasticity was described by the generalized Maxwell model composed of spring constants (elasticity) and dashpots (relaxation time). Relaxation time was obtained by fitting the modified Kelvin-Voigt model (spring and dashpot connected in parallel), assuming exponentially decaying force at a constant deformation ⁶⁴. Thus it permits establishing bulk elastic constant and relaxation time. The modified Kelvin-Voigt model can be described by the following:

$$F(t) = a\Delta x(1 - e^{-bt}), \text{ where } b = \frac{K_1+k_2}{n}, \text{ and } a = \frac{K_1k_2}{K_1+k_2}.$$

We focused on the part of a curve where the deformation (Δx) was kept constant and the force evolved as a negative exponential. We observed that our experimental data are best fitted by two different relaxation times, which could be described by the generalized Kelvin-Voigt model. The double exponential fit modeled our data better than a single exponential ⁶⁴, suggesting the presence of at least two processes behind the viscoelastic response. For more details on the model fitting, please consult the Matlab scripts at <https://github.com/inatamara/AFManalysisMatlab>.

a) water is not a physiological solution thus due to different osmolarity may destroy the tissue. In AFM studies, washing in water is not advisable as it may destroy the tissue and alter its mechanics. The results showing tissue treated with and without water should be presented.

We only used 1X PBS solution. Please see the 'Sample preparation for AFM' section, where this is clearly stated that all measurements were conducted in PBS to avoid sample dehydration with respect to physiological conditions (L508-511).

Water was only used to rinse OCT from tissue sections prior to reequilibration of the tissue charges in PBS and all measurements were conducted in physiological solution (PBS). A similar technique has previously been used by other groups (please see: Archterberg et al. J Invest Dermatol, 2014).

Our sample preparation method did not have any negative impact on tissue structure or integrity. Even after performing elasticity and viscoelasticity measurements on different areas of the same sample, the tissue was still intact, as demonstrated by hematoxylin & eosin staining:

Observations of 50 μm ovarian tissue sections under brightfield microscope, following AFM measurements. Arrows points toward follicles with intact structure and morphology. Scale 50 μm .

b) the authors present the nominal spring value - what was a variability of the cantilever spring constant. How many cantilevers were used in the measurements?

Force indentation curves were collected using a rectangular silicon nitride cantilever with a 0.07 N/m nominal spring constant and a 5- μm -diameter borosilicate glass spherical particle attached to the tip with a nominal spring constant of 0.07 N/m. The measured spring constant was very close to a nominal value of 0.07 ± 0.005 N/m using the thermal tune method.

We used only one cantilever for all experiments. Our AFM was thermally (and electrically) isolated, hence the variability of determining a spring constant is low.

c) did Dimitriadis correction was applied?

Calibration was done in air and tissue measurements in 1X PBS solution, so the Dimitriadis correction was not necessary.

d) what was the indentation dept analyzed.

The maximum indentation was set to ~ 1 μm . In order to apply a Hertzian model, indentation did not exceed half the radius of a bead (1.25 μm).

e) force curves should be shown together with moduli distributions, especially that distributions are not symmetric and using means +/- standard deviations is not the best way. Median is better.

The plot was easily corrected, as shown below.

P38:

Boxplots display 25th and 75th percentile, median (blue circle), and the whiskers extending to the last data point not considered outlier.

Mean was calculated from all force curves or force maps?

We selected force map areas based on the topology to avoid measurements of damaged surfaces or topologically variable (such as sudden jump) regions. Means were calculated on force-indentation curve bases.

f) how many maps were recorded? It seems that 3 per each tissue - this is not sufficient. What is the heterogeneity of an individual sample?

We did indeed measure at least three maps per sample. However, the the area was 100 x 100 μm^2 , which is much larger than in the previously reported paper suggested by the reviewer (see Deptula et al. ACS Biomater Sci Eng, 2020).

Measurement of each sample was limited to 40 min of the total experiment (from sample preparation) in order to limit sample damage, which explains why it was restricted to a restrained number of force maps (followed by viscoelasticity measurements). For a 100 x 100 μm square area, we performed 50 x 50 measurements, resulting in 2,500 force-indentation experiments. Each force-indentation experiment was treated with a Hertzian indentation model to extrapolate the apparent Young's modulus, with each pixel in a stiffness map representing Young's modulus from one force-indentation point. The scale is larger than tissue variability, so region-to-region variability within the same sample was low. This scale is important to even out topologically induced variability.

Here is the comparison of medians calculated over different regions in the same sample for different patients within each group.

Boxplots display 25th and 75th percentile, median, and the whiskers extending to the last data point not considered outlier.

g) why relaxation times are shown without error? They display mean or median?

Since we show full distribution (violin plot), error bars are not needed. In addition, our distributions are non-normal. The shown statistics display mean (line) and median (dot).

We can also show the overlay of violin plots with boxplots displaying 25th and 75th percentile, median (blue circle), and the whiskers extending to the last data point not considered outlier.

P38:

h) How many curves/maps were recorded to assess the viscoelasticity?

At least six measurements were conducted on at least nine different points in a sample. Viscoelasticity was measured in the same region that was used for force maps. We chose topologically stable regions.

f) what is the physical meaning of fast "tau" and slow "tau". What model was used to determine these values?

We have provided the following information in the new version of the materials and methods section:

L566-593:

To measure the viscoelasticity of ovarian tissue, we conducted repeated and successive long indentation cycles, followed by partial force release. For indentation portion, force was kept constant, while for the release portion, deformation was constant, allowing us to monitor the evolution of both deformation and force. The viscoelastic measurement (force spectroscopy) immediately followed elastic measurement (force mapping scan) and was performed in two points of the same tested region according to a force ramp design loop of ~240 s duration: cantilever extension with force increase (4- μ m indentation, 100 ms); constant height maintenance at the contact point (20 s); cantilever retraction and force decrease (0.8 μ m, 20 ms); and finally constant force maintenance (20 s). Data sampling frequency was set to 4,000 Hz.

Viscoelastic materials retain their shape after deformation, but with a time delay, as shown by the relaxation constant. In slow deformations, we can assume that the ECM is incompressible (Young modulus measurements). However, fast viscoelasticity measurements demonstrate that the tissue may be (reversibly) compressible in short time scales. In this context, viscoelasticity was described by the generalized Maxwell model composed of spring constants (elasticity) and dashpots (relaxation time). Relaxation time was obtained by fitting the modified Kelvin-Voigt model (spring and dashpot connected in parallel), assuming

exponentially decaying force at a constant deformation (Peaucelle et al. Curr Biol, 2011). Thus it permits establishing bulk elastic constant and relaxation time. The modified Kelvin-Voigt model can be described by the following:

$$F(t) = a\Delta x(1 - e^{-bt}), \text{ where } b = \frac{K_1+k_2}{n}, \text{ and } a = \frac{K_1k_2}{K_1+k_2}.$$

We focused on the part of a curve where the deformation (Δx) was kept constant and the force evolved as a negative exponential. We observed that our experimental data are best fitted by two different relaxation times, which could be described by the generalized Kelvin-Voigt model. The double exponential fit modeled our data better than a single exponential (Peaucelle et al. Curr Biol, 2011), suggesting the presence of at least two processes behind the viscoelastic response. For more details on data processing and the model fitting, please consult the Matlab scripts at <https://github.com/inatamara/AFMAnalysisMatlab>.

L1024-1030:

8) Roughness should be correlated with SEM images. How many images were used to calculate S_a value? What is the importance of using 10 $\mu\text{m} \times 10 \mu\text{m}$ topography images

(and roughness) as compared to SEM images that better demonstrate the age-dependent difference in ovary tissue?

Roughness measurements are directly extracted from SEM images captured at 12,000X magnification (image size without cropping: 10 μm x 10 μm), which is an intermediate value between 5,000X (fiber level) and 20,000X (fiber bundle level). This allows us to explore the surface topography that can be sensed by ovarian cells at the cell scale, and directly reflects the topography of the ECM fibrous network observed and analyzed by SEM.

Contrary to profilometry analyses, our technique directly links architecture to roughness. The technique is described in the 'Topography measurement' section in the materials and methods and the workflow has been summarized in Fig. 6D.

For clarity, we have appended the SEM images and stereoscopic construction of each 3D model used in Figure 6. Purple and blue cross overlapping demonstrates the eucentricity of SEM image tilting.

P42:

9) The authors do not cite any tissue-oriented AFM studies. This is important to evaluate the quality of the obtained results in terms of tissue mechanics. This unfortunately explain weak points in AFM-based analysis of ovarian tissue. Some references:

Puttini et al. Mol. Therapy 2009 - muscle tissue

Plodinec et al. Nature Nanotechnol 2012 - breast cancer

Lekka et al. Arch. Biochem.Biophys 2012 - breast, vulvar, endometrium

Tan et al. 2015 Nanoscale - liver tissue

Bouchonville et al. Soft matter 2016 - brain tissue

Ciasca et al. Nanoscale 2016 - brain tissue

Anura et al. J. mech. Behav. Biomed. Devices 2017 - epithelial connective tissue

Deptula et al. ACS Biomater. Sci. Eng. 2020 - colon cancer

We would like to thank Reviewer #3 for the suggestions. The references can be easily included in the manuscript.

We would like to stress that one co-author, Dr. Alexis Peaucelle, is a pioneer in tissue-based rheological and mechanical AFM measurements. He has published both in plant

science (please see Peaucelle et al. *Curr Biol*, 2011; Uytewaal et al. *Cell*, 2012; Braybrook et al. *Plos One*, 2013, Peaucelle et al. *JOVE*, 2014, Peaucelle et al. *Curr Biol*, 2015; Feng et al. *Curr Biol*, 2018, Alonso-Serra et al. *Curr Biol*, 2020; Sampathkumar et al. *Development*, 2019, Peaucelle et al. *Contr Méchanique*, 2020) and animal tissue (Fleury et al. *Phys Rev*, 2016; Fleury et al. *Bioarxiv*, 2020).

We regret that we did not provide a detailed methods section, which made this reviewer think that we are novices in AFM-based measurement of tissue mechanics. However, as is clear from our previous replies, this was easily corrected in the new version of the manuscript.

Calibration is well described in Schillers et al. *Sci. Reports* 2017.

In the previous version of our manuscript, we already described the calibration method. Please see below:

'Prior to each experiment, the spring constant of the probe was determined by thermal fluctuation force calibration in air using a glass slide.'

Please note that we used this method in all our previously published studies (as outlined above).

Reviewers' Comments:

Reviewer #1:

Remarks to the Author:

The revised manuscript by Dr. Amorim and colleagues sufficiently addressed the concerns of this reviewer, and addressed the issues raised by all reviewers to my satisfaction.

In particular, the questions raised about menstrual cycle phase (proliferative versus secretory or menstrual timing) are unlikely to negate the important findings presented. In my view, additional experimental data is beyond the scope of this work. The revised figures and explanations provide sufficient evidence for readers to rigorously interpret the data presented and the strength of evidence is compelling.

The revised manuscript represents an important step forward toward understanding the enigma of primordial follicle arrest and female gametogenesis in the ovary. It is a contribution that is certain to be cited widely in the field of reproduction and it provides a framework for future research.

Reviewer #3:

Remarks to the Author:

The authors extensively revised the manuscript answering all my doubts. The manuscript escapes from presenting only a description of the study into that showing better the biological significance of the presented work.

Studying plant tissues by AFM helps to make measurements with human tissues. However, each sample has its own specificity also in tissue mechanics. The impression of being a novice in AFM stems from weak AFM description and not an exact description of the experiments. An example is below:

Dimitriadis correction is not linked with the calibration (as the authors stated in the response) but it accounts for the correction of Young's modulus of biological samples affected by the presence of stiff underlying substrate. The authors cite the correct paper (it is about the correction of Young's modulus, no calibration). Currently, there is an ongoing discussion among AFM groups guided by renowned scientists (pioneers of cells and tissues mechanics) that finite thickness correction has to be applied, even when pyramidal probes are used. In the presented manuscript (although not highlighted by the authors), the sample height seems to be too large as compared to indentation depth, however, a spherical probe was used. The effect of underlying stiff substrate is enormous for such probes and could be visible also for thick samples.

Despite my criticism, being honest, I say that now I see that the manuscript present the work with a better explanation of its significance and meets the requirements of good scientific work.

Therefore, I recommend the paper to be accepted for NCOMMS publication.